# Hedgehog signaling is required for endomesodermal patterning and germ cell development in the sea anemone *Nematostella vectensis*

Cheng-Yi Chen[1], Sean A McKinney[1], Lacey R Ellington[1], Matthew C Gibson[1,2]*

[1]Stowers Institute for Medical Research, Kansas City, United States; [2]Department of Anatomy and Cell Biology, The University of Kansas School of Medicine, Kansas City, United States

**Abstract** Two distinct mechanisms for primordial germ cell (PGC) specification are observed within Bilatera: early determination by maternal factors or late induction by zygotic cues. Here we investigate the molecular basis for PGC specification in *Nematostella*, a representative pre-bilaterian animal where PGCs arise as paired endomesodermal cell clusters during early development. We first present evidence that the putative PGCs delaminate from the endomesoderm upon feeding, migrate into the gonad primordia, and mature into germ cells. We then show that the PGC clusters arise at the interface between *hedgehog1* and *patched* domains in the developing mesenteries and use gene knockdown, knockout and inhibitor experiments to demonstrate that Hh signaling is required for both PGC specification and general endomesodermal patterning. These results provide evidence that the *Nematostella* germline is specified by inductive signals rather than maternal factors, and support the existence of zygotically-induced PGCs in the eumetazoan common ancestor.

*For correspondence:
mg2@stowers.org

**Competing interests:** The authors declare that no competing interests exist.

## Introduction

During development, animal embryos typically set aside a group of primordial germ cells (PGCs) that later mature into germline stem cells (GSCs) and in turn give rise to gametes during adulthood (*Nieuwkoop and Sutasurya, 1979*; *Nieuwkoop and Sutasurya, 1981*; *Wylie, 1999*; *Juliano et al., 2010*). The process of PGC specification both underpins the sexual reproduction cycle and involves transitions of pluripotency, making the mechanisms that distinguish germ cells from soma of critical importance in developmental and stem cell biology (*Solana, 2013*; *Irie et al., 2014*; *Magnúsdóttir and Surani, 2014*). PGC specification occurs early in development and could hypothetically rely on autonomous or non-autonomous cues. Historically, comparative studies of germ cell development defined the core mechanisms of PGC specification as preformation and epigenesis (*Nieuwkoop and Sutasurya, 1979*; *Nieuwkoop and Sutasurya, 1981*; *Extavour and Akam, 2003*). For clarity, here we adopt the terms 'inherited' and 'induced' to distinguish between PGC specification mechanisms (*Seydoux and Braun, 2006*; and *Seervai and Wessel, 2013*). In inherited PGC specification (e.g. *Drosophila*, *C. elegans* and *Danio rerio*), cytoplasmic determinants referred to as the germ plasm are maternally deposited in oocytes and then segregated into specific blastomeres through cell division (*Strome and Wood, 1982*; *Williamson and Lehmann, 1996*; *Yoon et al., 1997*). In contrast, there are neither maternal germline determinants nor pre-determined PGC fates in specific blastomeres in inductive PGC specification. For example, BMP signaling is required for PGC specification from precursor cells in mouse, axolotl, and cricket embryos (*Lawson et al., 1999*; *Chatfield et al., 2014*; *Nakamura and Extavour, 2016*). The inductive mode of PGC specification is

more prevalent across the animal kingdom, and therefore hypothesized to reflect mechanisms present in the cnidarian-bilaterian common ancestor (*Extavour and Akam, 2003*).

Strictly categorizing PGC specification mechanisms into one of two types may fail to account for the true variety present in nature. Highly regenerative animals such as sponge, Hydra and planaria maintain multipotent stem cells that are capable of differentiating into both soma and germ line in adults (*Bosch and David, 1987*; *Fierro-Constaín et al., 2017*; *Issigonis and Newmark, 2019*), thus representing an alternative post-embryonic mode of PGC specification. In sea urchin, a combination of inherited and inductive PGC specification mechanisms are observed (*Voronina et al., 2008*). It follows that the germline determination mechanisms of a wide variety of organisms may lie at different positions along the continuum between maternal inheritance and zygotic induction (*Nieuwkoop and Sutasurya, 1981*; *Seervai and Wessel, 2013*).

Cnidarians (jellyfish, sea anemones and corals) are the sister group to bilaterians and occupy an ideal phylogenetic position for investigating likely developmental traits of the eumetazoan common ancestor (*Technau and Steele, 2011*; *Russell et al., 2017*). Among cnidarians, the sea anemone *Nematostella vectensis* maintains distinct adult gonad tissue and features PGC specification dynamics hypothesized to reflect an evolutionary transition from induction to inherited mechanism based on expression patterns of conserved germline genes (*Extavour et al., 2005*). Additionally, a well-annotated genome (*Putnam et al., 2007*), defined developmental stages (*Fritzenwanker et al., 2007*) and diverse genetic tools (*Ikmi et al., 2014*; *Renfer and Technau, 2017*; *He et al., 2018*; *Karabulut et al., 2019*) make *Nematostella* a genetically tractable model to elucidate developmental mechanisms controlling PGC specification.

In this study, we explore mechanisms of PGC development in *Nematostella* and test whether the putative PGC clusters are specified by maternal or zygotic control. We first follow the development of putative PGCs and provide evidence supporting their germ cell fate in adults. We then leverage shRNA knockdown and CRISPR/Cas9 mutagenesis to interrogate the developmental requirements for the Hedgehog signaling pathway in PGC specification. From these results, we conclude that Hh signaling is either directly or indirectly required for PGC specification in *Nematostella*. As Hh signaling is only activated zygotically, these data indicate an inductive mechanism for *Nematostella* PGC specification and support the inference that the eumetazoan common ancestor likely specified PGCs via zygotic induction.

## Results

### Evidence that PGCs form in primary polyps and migrate to gonad rudiments

The localized expression of the conserved germline genes *vasa* and *piwi* suggest that *Nematostella* PGCs arise during the metamorphosis of planula larvae into primary polyps between 4 and 8 days-post-fertilization (dpf; *Figure 1A–D''*; *Extavour et al., 2005*; *Praher et al., 2017*). These cells first appear as discrete clusters within the two primary mesenteries, in close proximity to the pharynx (*Figure 1B–D''*). To follow the development of putative PGCs at higher spatio-temporal resolution, we generated a polyclonal antibody against *Nematostella* Vasa2 (Vas2) and used immunohistochemistry and fluorescent in situ hybridization to confirm that Vas2 was co-expressed with *piwi1* and *piwi2* in the putative PGC clusters (*Figure 1—figure supplement 1A–L*). Further supporting their germline identity, we also found that *tudor* was enriched in putative PGC clusters (*Figure 1—figure supplement 1M–P*; *Boswell and Mahowald, 1985*; *Arkov et al., 2006*; *Chuma et al., 2006*). We next quantified the number of putative PGCs in primary polyps and found that Vas2+ cell numbers varied between individuals, with a median number of 10 cells per primary polyp (*Figure 1E*). We did not observe a significant difference in Vas2+ cell numbers between two primary mesenteries (*Figure 1—figure supplement 2*). Within the same spawning batch, there was no significant difference in the number of putative PGCs in primary polyps assayed on different days. This suggests that there was neither loss nor expansion of PGCs in primary polyps prior to feeding and development of the juvenile stage.

Adult *Nematostella* harbor mature gonads in all eight internal mesentery structures while confocal images found that Vas2+ epithelial cell clusters were localized only at endomesodermal septa associated with segments s2 and s8 (*Figure 2A–B'*; *Video 1*; *Figure 2—figure supplement 1*;

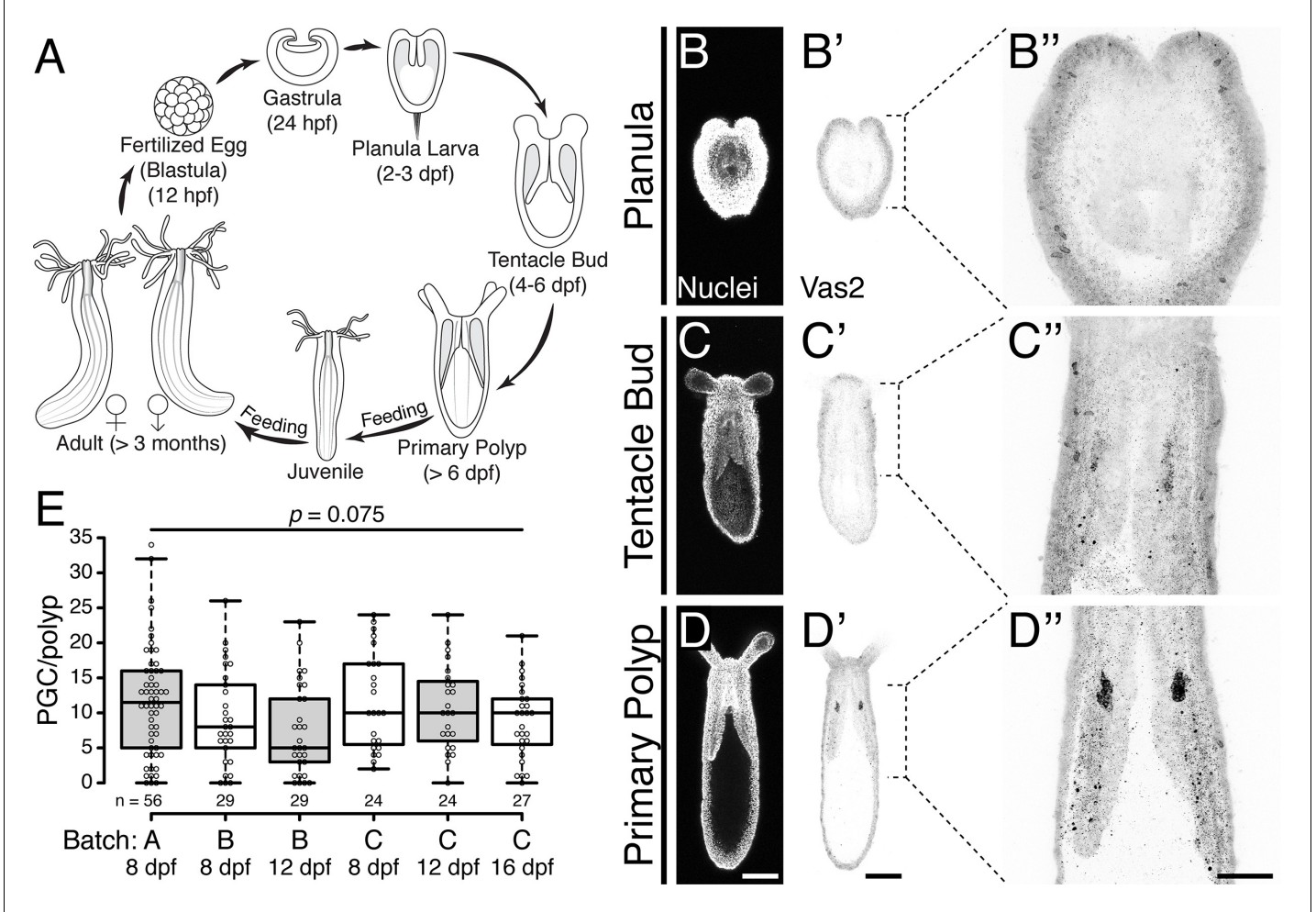

**Figure 1.** Putative *Nematostella* PGCs clusters are specified during metamorphosis. (**A**) Diagram depicting the *Nematostella* life cycle. (**B–D''**) During the transition from tentacle bud stage larvae to metamorphic primary polyps, Vas2 expression is gradually enriched in the primary mesenteries in two endomesodermal cell clusters in proximity to the pharynx. (**E**) Quantification of PGC numbers on 8, 12 or 16 dpf from three spawning batches. ANOVA analysis did not show significant differences among mean PGC number in primary polyps on different days post-fertilization. Center lines show the medians; box limits indicate the 25th and 75th percentiles; whiskers extend 1.5 times the interquartile range from the 25th and 75th percentiles; data points are plotted as open circles. Scale bar = 100 μm in **D** and **D'**; 50 μm in **D''**. **B–D'** are at the same scale; **B''–D''** are at the same scale.

The online version of this article includes the following source data and figure supplement(s) for figure 1:

**Source data 1.** PGC numbers on 8, 12 or 16 dpf from three spawning batches.
**Figure supplement 1.** Conserved germline-related genes are expressed in putative PGC clusters.
**Figure supplement 2.** PGC numbers in the segment 2- and segment 8-associated primary mesenteries at three different timepoints post fertilization.
**Figure supplement 2—source data 1.** PGC numbers in the segment 2- and segment 8-associated primary mesenteries at three different timepoints post fertilization.

*Williams, 1975*; *Frank and Bleakney, 1976*; *He et al., 2018*). If the two Vas2+ epithelial cell clusters are the only precursors for adult germ cells, it follows that these cells would have to delaminate and migrate to populate the eight gonad rudiments. Alternatively, new PGCs could arise within each of the six non-primary mesenteries, perhaps at a later developmental stage. To distinguish between these possibilities, we examined the localization of Vas2-expressing putative PGCs in primary polyps and later juvenile stages. In all primary polyps, putative PGCs appeared in two coherent clusters at 8 dpf (*Figure 2B–B'*). In older primary polyps (>10 dpf), some PGC clusters cells appeared to stretch basally through the underlying mesoglea, an elastic extracellular matrix that separates epidermis and gastrodermis layers (*Figure 2C–C'*; *Schmid, 1991*; *Shaposhnikova et al., 2005*). After feeding for more than a week, primary polyps start adding tentacles and enter the juvenile stage.

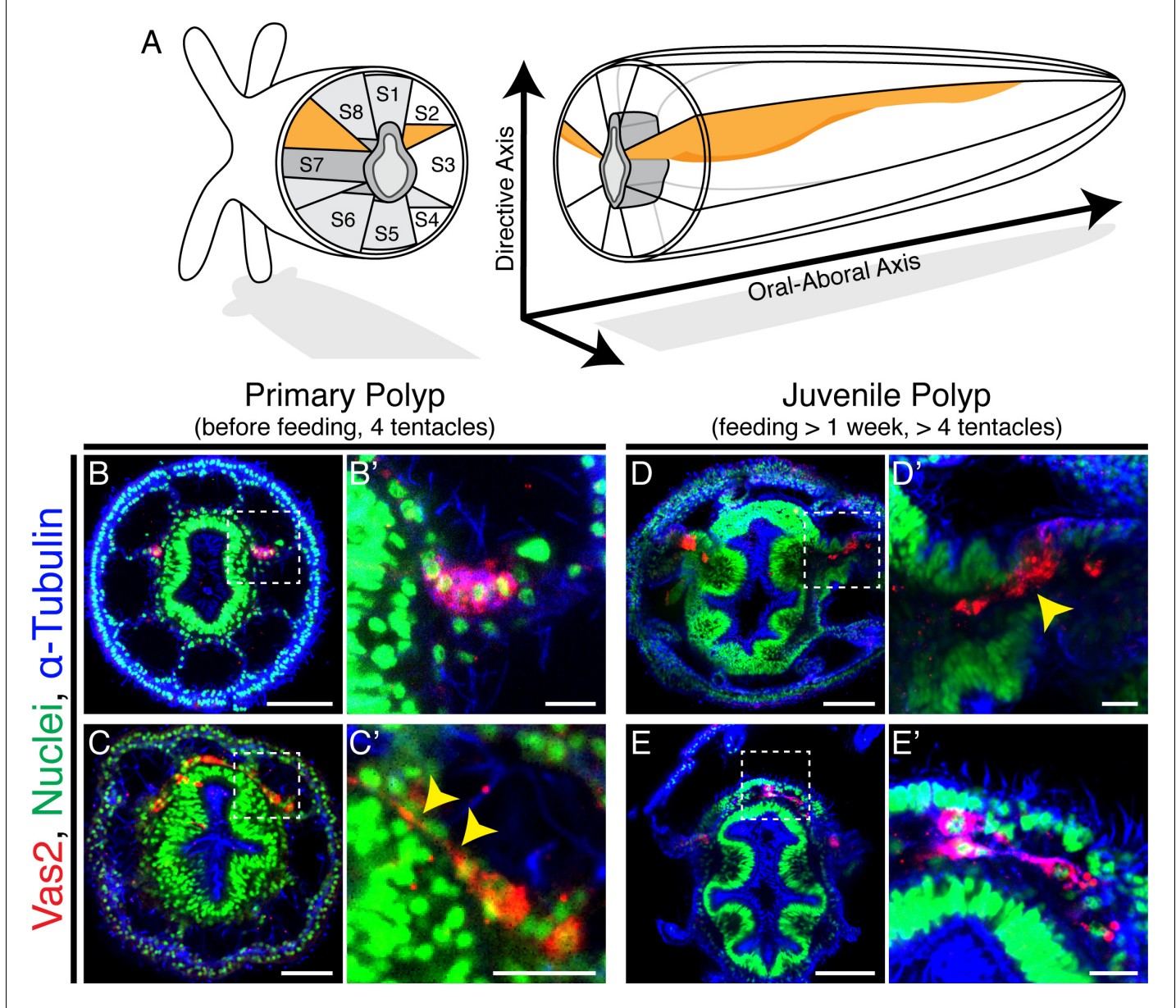

**Figure 2.** Putative *Nematostella* PGCs delaminate through epithelial-mesenchymal transition (EMT) and appear to migrate to non-primary mesenteries. (A) Schematic diagram of *Nematostella* polyp anatomy depicts the pharynx and mesentery arrangements at the pharyngeal level. The eight mesenteries (two primary mesenteries in *orange* and six non-primary mesenteries in *light gray*) harbor gonad epithelium, muscle and digestive tissue. The internal structures of *Nematostella* are arrayed around the pharynx (*dark gray*). The *Directive* and *Oral-Aboral* axes are indicated. Segment nomenclatures follow **He et al., 2018**. (B–B') Paired clusters of putative PGCs labeled by Vas2 immunofluorescence (*red*) initially exhibit epithelial charateristics. (C–C') Putative PGCs from >10 dpf primary polyps appear to stretch their cell bodies basally (*yellow* arrowheads). (D–D') Following nutrient intake, putative PGCs delaminate into the mesoglea through an apparent EMT (*yellow* arrowhead). (E–E') In the mesoglea, these Vas2+ cells exhibit fibroblast-like morphology and are detected between mesenteries at the level of the aboral pharynx. Scale bar = 50 μm in B, D, and E; 20 μm in C; 10 μm in B', C', D', and E'.

The online version of this article includes the following figure supplement(s) for figure 2:

**Figure supplement 1.** Development of oocytes or sperm in the mesenteries of female (A–A') and male (B–B') adult polyps.
**Figure supplement 2.** Migratory PGCs show fibroblast-like morphology and pseudopodia.
**Figure supplement 3.** *twist* expression in putative PGCs suggests their migratory potential.

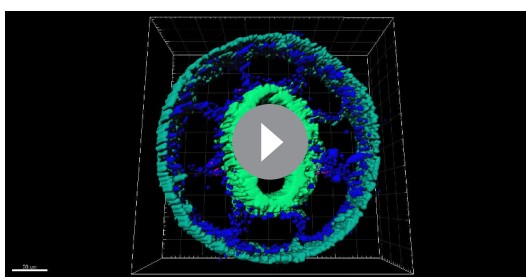

**Video 1.** 3D reconstruction of pharyngeal structures. PGCs (*magenta*) are specified on the epithelium of the two primary mesenteries, close to the pharynx (*cyan* cells in the center). The endomesodermal nuclei are pseudocolored in *blue*, and the ectodermal nuclei are in *cyan*.
https://elifesciences.org/articles/54573#video1

Interestingly, upon feeding, Vas2-expressing putative PGCs appeared to delaminate from the epidermis into the underlying mesoglea (*Figure 2D–D'*). Putatively delaminated Vas2 positive cells displayed a fibroblast-like morphology with pseudopodial protrusions, similar to other migratory cell types (*Figure 2—figure supplement 2*; *Scarpa and Mayor, 2016*). Further consistent with migratory potential, Vas2+ cells also expressed *twist* (*Figure 2—figure supplement 3*), a conserved regulator of mesoderm development and a marker of metastatic cancer cells (*Yang et al., 2004*; *Kallergi et al., 2011*). While specification of additional PGC clusters was not observed on the six non-primary mesenteries, we did find evidence for a process of radial cell migration between mesenteries at the level of the aboral end of the pharynx, where the mesoglea between ectoderm and endomesoderm increases in volume after the primary polyp stage (*Figure 2E–E'*).

We next followed the localization of putative PGCs through successive developmental timepoints. The majority of 10 dpf primary polyps showed PGCs within clusters (*Figure 3A–A'*), while >10 dpf primary polyps showed some PGCs localized between the primary mesenteries and segment s1 (*Figure 3B–B'*). The direction of this initial migration toward segment s1 suggests the existence of attractive/repulsive signals for migratory PGCs. Additionally, we found that in juveniles putative PGCs migrated aborally toward the mesentery region where gonad rudiments will mature in adults (*Figure 3C–D'*). These migratory PGCs were proliferative as shown by Phospho-Histone H3 labeling and EdU incorporation (*Figure 3—figure supplement 1*). Combining all observations, we hypothesize that in *Nematostella*, putative PGCs initially form in primary polyps as two endomesodermal cell clusters at the level of the aboral pharynx. During juvenile stages, we further postulate that these cells delaminate into the mesoglea layer between the ectoderm and endomesoderm via an apparent epithelial-mesenchymal transition (EMT) and then migrate to the gonad rudiments.

## Evidence that putative PGCs mature and give rise to germ cells in adult gonads

To assess the germline identity of putative PGCs, we next followed the development of Vas2+ cells from juvenile to young adult stages (>2 month-old). In maturing polyps, the endomesodermal mesenteries are organized from proximal (external) to distal (internal) into parietal muscles, retractor muscles, gonads and septal filaments, with occasionally observed ciliated tracts between the gonads and septal filaments (*Figure 4A–D*; *Williams, 1979*; *Jahnel et al., 2014*). In juvenile polyps, Vas2+ cells were observed in the mesoglea between the septal filaments and the retractor muscles (*Figure 4C*), an endomesodermal region that will later form the adult gonad epidermis. We also occasionally observed putative PGCs between the ciliated tracts and the retractor muscles (*Figure 4D*). After feeding for 8 weeks, most polyps reached the 12-tentacle stage and the mesenteries progressively matured, becoming wider and thicker. At this stage we observed Vas2+ putative PGCs in the maturing gonad region, along with Vas2+ immature oocytes or sperm cysts in females and males, respectively (*Figure 4E–F*). In adult mesenteries, similar putative PGCs were found in the vicinity of oocytes and sperm stem cells (*Figure 4—figure supplement 1A–C'*) and in the aboral region of mesenteries (*Figure 4—figure supplement 1D–E'*, and summarized in *Figure 4—figure supplement 1F*). In the aboral regions, Vas2+ PGC-like cells exhibited proliferative capacity as evidenced by EdU incoporation (*Figure 4—figure supplement 2*). Taken together, these observations suggest that the putative PGCs comprise a continuous Vas2-expressing lineage that proliferates and ultimately gives rise to mature germ cells. As proposed by previous work (*Extavour et al., 2005*), these data support the hypothesis that the germline gene-expressing cell clusters of primary polyps represent *bona fide* PGCs of *Nematostella*.

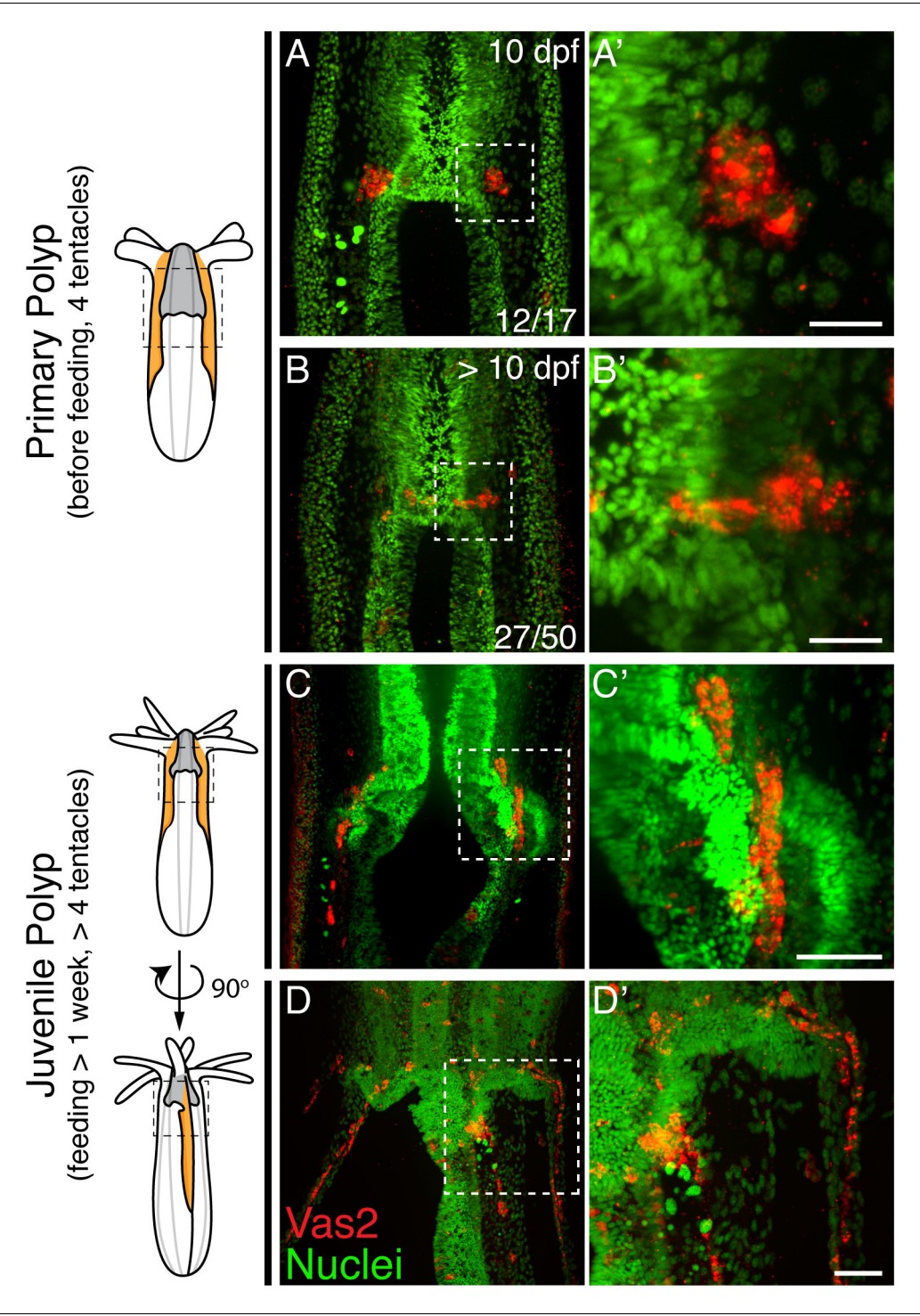

**Figure 3.** *Nematostella* PGCs migrate aborally to the gonad rudiments during the juvenile stage. (**A–A'**) The majority of young primary polyps (≤10 dpf) exhibit two PGC clusters (Vas2+, *red*) in close proximity to the pharynx. (**B–B'**) In more mature primary polyps (>10 dpf), some PGCs elongate and localize between the mesenteries. (**C–C'**) Following feeding, putative PGCs spread aborally into the gonad rudiments. (**D–D'**) A juvenile polyp viewed 90 degrees from the orientation of **C**, showing aborally migrating PGCs in non-primary mesenteries. Scale bar = 10 µm in **A'** and **B'**; 20 µm in **C'** and **D'**.

The online version of this article includes the following figure supplement(s) for figure 3:

**Figure supplement 1.** PGCs are proliferative in juvenile polyps.

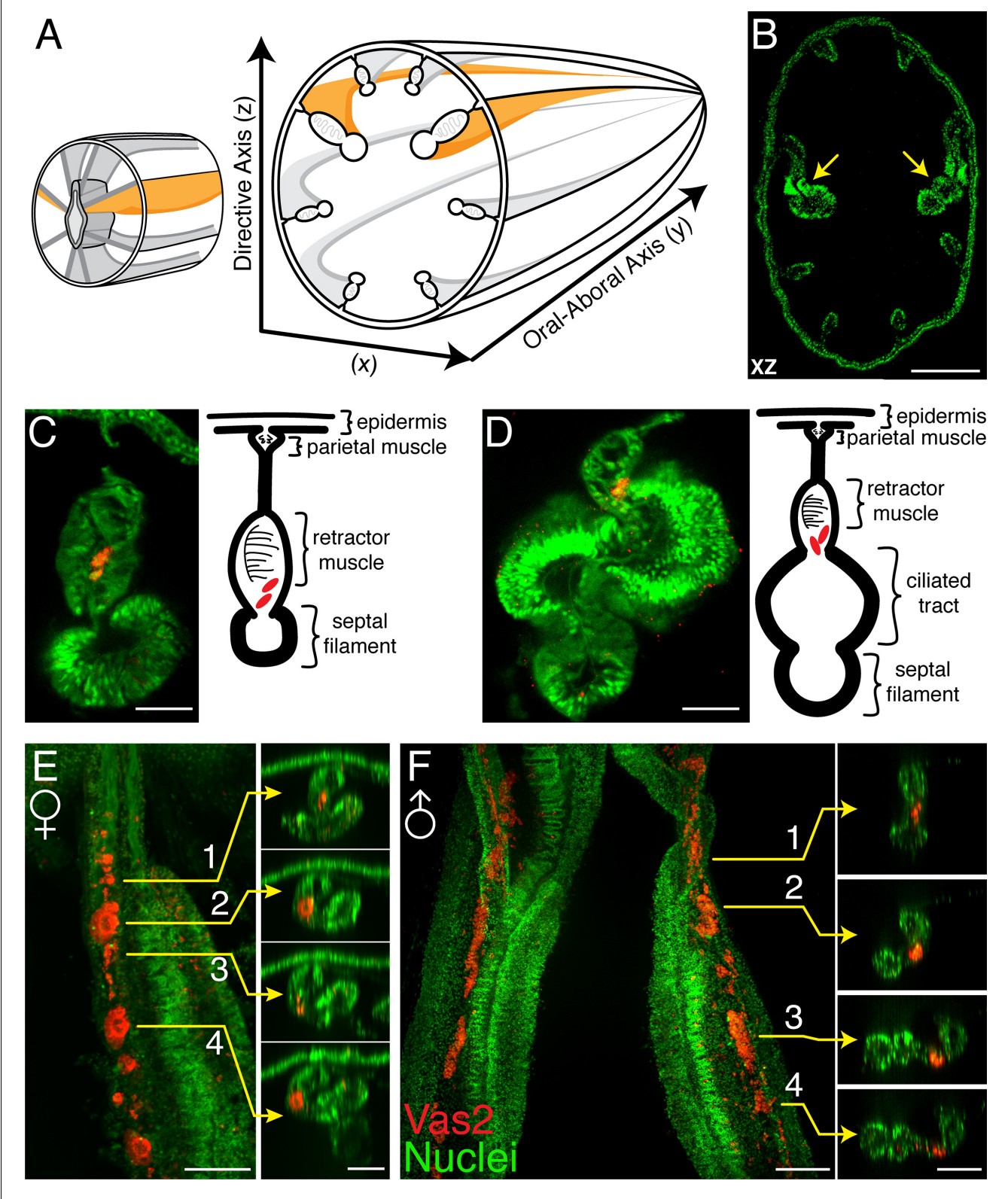

**Figure 4.** Vas2+ germ cells in juvenile gonad rudiments. (**A**) Schematic diagram of *Nematostella* polyp anatomy depicts the gametogenic mesenteries at the mid-body level. (**B**) A mid-body level cross section through a juvenile polyp, note the enlarged primary mesenteries (*yellow* arrows). Nuclei are counter stained by DAPI (*green*). (**C and D**) Representative images of maturing mesenteries with corresponding schematic diagrams. Putative PGCs are labeled by Vas2 immunofluorescence in *red*. (**E**) Whole-mount juvenile female mesentery shows Vas2-labeled putative PGCs and germ cells in close

*Figure 4 continued on next page*

*Figure 4 continued*

proximity (*red*), suggesting maturing oocytes originate from the continuous PGC lineage. (F) Whole-mount juvenile male mesentery shows Vas2-labeled putative PGC and germ cell populations, including the rudimentary sperm cysts. Insets 1–4 of E and F are xz plane views at the indicated levels. Scale bar = 100 µm in B; 20 µm in C–F.

The online version of this article includes the following figure supplement(s) for figure 4:

**Figure supplement 1.** Adult PGC-like lineages localize adjacent to the mature gonad and migrate aborally.

**Figure supplement 2.** PGC-like cells are proliferative.

## Evidence supporting a zygotic mechanism for primordial germ cell specification

We next investigated whether *Nematostella* PGCs are specified by inheritance of maternal determinants or through induction by zygotically-expressed factors. In the inheritance-based mechanisms of other species, maternally-deposited germline determinants are segregated into specific PGC precursors during early cleavage (*Nieuwkoop and Sutasurya, 1979*; *Nieuwkoop and Sutasurya, 1981*; *Extavour and Akam, 2003*). In *Nematostella*, prior to the appearance of putative PGC clusters in developing polyps, we observed perinuclear Vas2 granules that could hypothetically serve as maternal germline determinants (*Figure 5*). These granules were previously identified with an independent antibody and proposed to regulate *Nematostella* piRNAs (*Praher et al., 2017*). However, the perinuclear Vas2 granules were distributed homogenously around oocyte germinal vesicles (*Figure 5A*), in every cell of blastulae, and in most endomesodermal cells after gastrulation (*Figure 5B*; *Praher et al., 2017*). In endomesodermal cells, Vas2+ granules gradually diminished when the putative PGC cluster cells activated Vas2 expression (*Figure 5B–E''*), suggesting that germ cell fate was gradually specified in endomesodermal precursor cells rather than being maternally predetermined in a set of germline precursor cells. Furthermore, germline gene transcripts (i.e. *vas1*, *vas2*, *nos2* and *pl10*) displayed a homogenous distribution in the endomesoderm of embryos and larva before PGC specification (*Extavour et al., 2005*). Endomesodermal enrichment of germline genes before *Nematostella* PGC formation could be consistent with the proposed germline multipotency program (GMP), where the expression of conserved germline factors underlies the multipotency of progenitor cells (*Juliano et al., 2010*). In line with the GMP hypothesis, we hypothesized that *Nematostella* PGCs are specified from a pool of multipotent endomesodermal precursors as observed in other species that use zygotic mechanisms to induce PGCs.

In primary polyps, putative PGC clusters initially form in the two primary mesenteries, which are distinguished by the presence of aborally-extended regions of pharyngeal ectoderm known as septal filaments (*Figure 6A–A'*, *Video 1*; *Steinmetz et al., 2017*). While the mechanism of primary mesentery specification is unknown, this process likely lies downstream of Hox-dependent endomesodermal segmentation in developing larvae. Interestingly, segmentation of the presumptive primary mesenteries is disrupted in both *Anthox6a* mutants and *Gbx* shRNA-KD polyps (*He et al., 2018*). In both of these conditions, we observed aberrant attachment of the septal filaments and the associated induction of PGC clusters in non-primary septa (*Figure 6B–D'*). This suggests that the precise location of the putative PGC clusters can be subject to regulation, and hints at the existence of zygotic PGC-inducing signals from the pharyngeal ectoderm.

## PGC specification is dependent on zygotic Hedgehog signaling activity

Previous gene expression studies have suggested that the Hh signaling pathway may be involved in patterning the endomesoderm and potentially the formation of germ cells (*Matus et al., 2008*). Using double fluorescent in situ hybridization to detect the expression of *Nematostella hedgehog1* (*hh1*) and its receptor *patched* (*ptc*) in late planula larvae, we found that both ligand and receptor were expressed in reciprocal domains of ectoderm and endomesoderm associated with the pharynx (*Figure 7A*). Later, the PGC clusters appeared within the endomesodermal *ptc* expression domain, adjacent to where *hh1* is expressed in the pharyngeal ectoderm (*Figure 7B–D'*). Because PGCs formed in association with the juxtaposed *hh1* and *ptc* expression domains, we hypothesized that Hh signaling may direct neighboring endomesodermal cells to assume PGC identity (*Figure 7E*).

To test functional requirements for Hh signaling in *Nematostella* development, we used shRNA-mediated knockdown and CRISPR/Cas9-directed mutagenesis (*Ikmi et al., 2014*; *Kraus et al., 2016*;

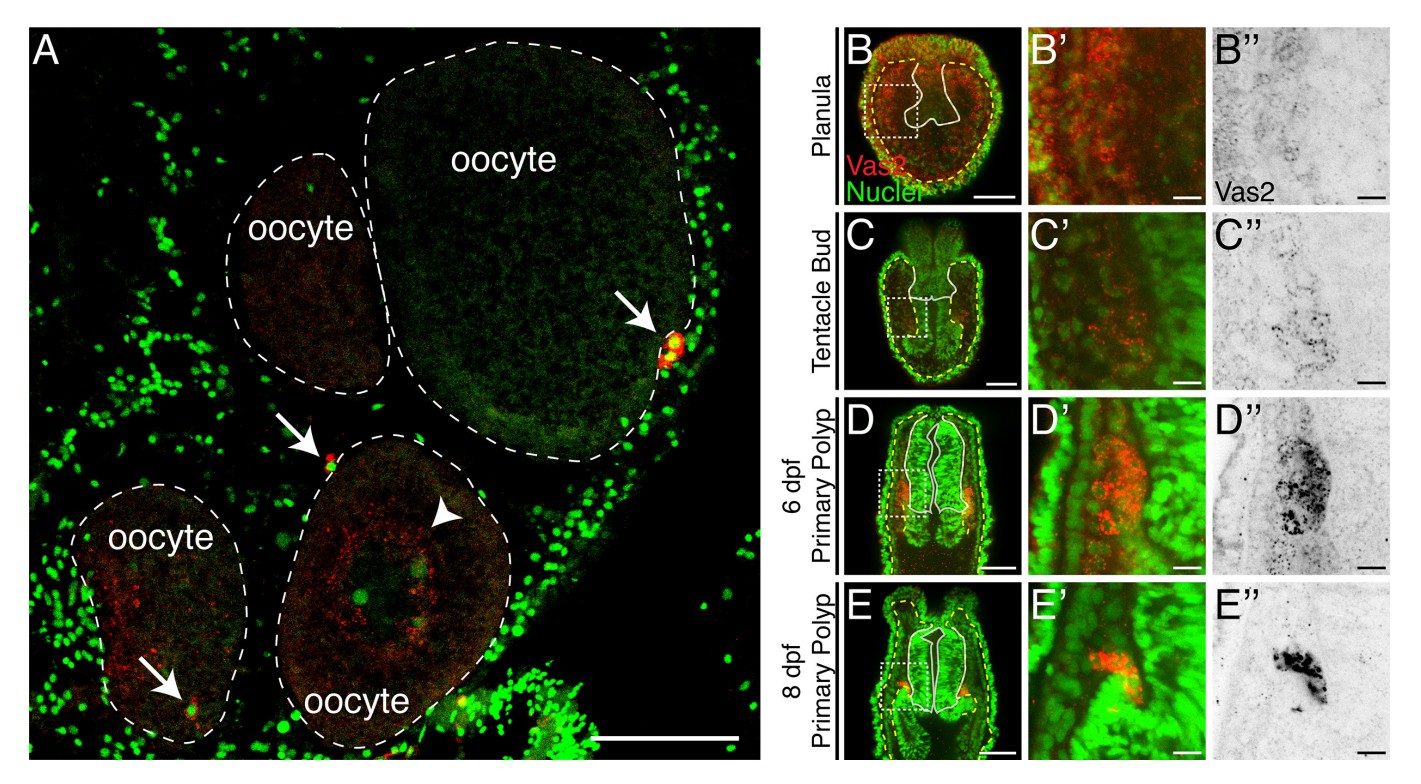

**Figure 5.** Maternally inherited perinuclear Vas2 granules diminish during PGC specification. (**A**) Cross-section of female gonad. Vas2 protein (*red*) forms puncta that surround the nuclei of maturing oocytes (arrowhead). In addition, PGC-like cells (arrows) are found next to oocytes. Oocyte boundaries are delineated by dotted lines (*white*). Nuclei and nucleolus are counter stained by Hoechst (*green*). (**B–E''**) Perinuclear Vas2 granules in endomesodermal cells gradually diminish as the PGC clusters are specified next to the pharynx during the tentacle bud through primary polyp stages. The pharynx is delineated by *white* lines and the boundary between ectoderm and endoderm is delineated by *yellow*-dotted lines. (**B'–E''**) Enlarged views of boxed areas in B-E. B-E'' are single focal planes under the same imaging conditions. Scale bar = 50 µm in **A–E**; 10 µm in **B'–E''**.

*He et al., 2018*). Unfertilized eggs were injected with shRNAs targeting either *hh1* or *gli* (a transcription factor downstream of Hh signaling) or with two independent gRNAs targeting *gli*. Using the expression of Vas2 protein and *piwi1* transcripts as readouts for PGC identity, we found that PGC specification was significantly inhibited in both knockdown (*Figure 7F–J*) and presumptive F0 mutant primary polyps (*Figure 7K–M*; see *Figure 7—figure supplement 1* and Materials and methods for *gli* gRNA target sites). These data suggest that the Hh signaling pathway is required for normal *Nematostella* PGC specification.

During Hh signal transduction, binding of Hh ligand to Ptc de-represses the transmembrane protein Smoothened (Smo), which in turn activates a cytoplasmic signaling cascade (*Forbes, 1993*; *Alcedo et al., 1996*; *Stone et al., 1996*; *van den Heuvel and Ingham, 1996*; *Bangs and Anderson, 2017*). To further test the involvement of Hh signaling in PGC formation, we treated developing animals with the Smo antagonists GDC-0449 (Vismodegib) or Cyclopamine (*McCabe and Leahy, 2015*; *Sharpe et al., 2015*). When early gastrulae were treated with either inhibitor, we did not observe significant developmental defects at our working concentration (*Figure 8A–D*). However, PGC numbers were significantly reduced (*Figure 8E–H*). To test Hh requirements for the establishment versus maintenance of PGC identity, we treated developing *Nematostella* with GDC-0449 either during PGC specification (4–8 dpf) or post-PGC specification (8–12 dpf; *Figure 8—figure supplement 1A*). Consistently, PGC formation was significantly inhibited by GDC-0449 treatment during 4 to 8 dpf metamorphosis (*Figure 8—figure supplement 1B*, compare Ctrl and GDC). In contrast, PGC number showed no significant difference when the pathway was inhibited after specification (*Figure 8—figure supplement 1B*, compare Ctrl-Ctrl and Ctrl-GDC). However, we observed significantly fewer

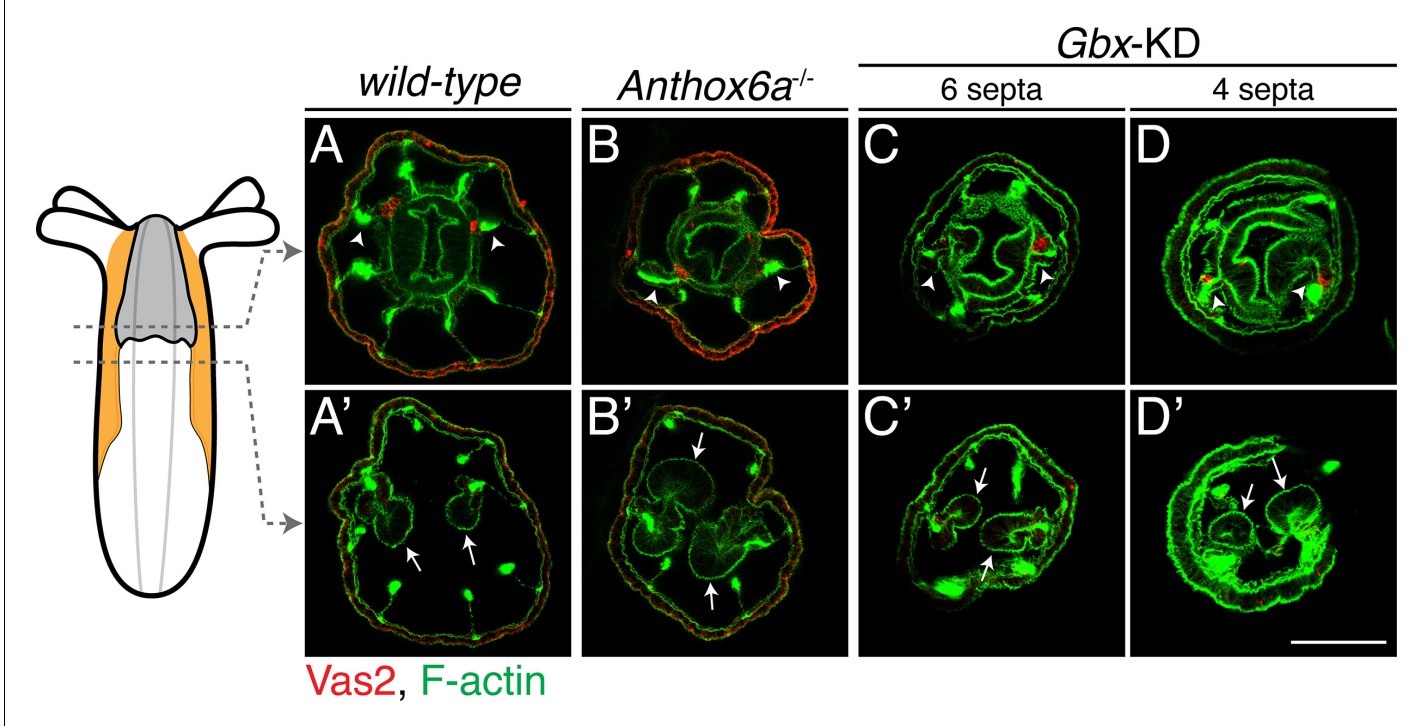

**Figure 6.** PGC clusters are specified on mesenteries with primary septal filaments of *wild-type*, *Anthox6a* mutant and *Gbx* shRNA knockdown primary polyps. Although primary mesenteries are missing in *Anthox6a* mutants or *Gbx* shRNA knockdown primary polyps, PGCs (labeled by Vas2 immunofluorescence in *red*) are specified on the mesenteries (arrowheads) where primary septal filaments attach (arrows). Scale bar = 50 μm in **D'**. All images are at the same scale.

PGCs in long-term control treatments (0.05% DMSO between 4–12 dpf, Ctrl-Ctrl) than in short-term controls (4–8 dpf, Ctrl). In *wild-type* primary polyps, there was no significant difference in mean PGC number between 8, 12 and 16 dpf primary polyps (*Figure 1E*). We therefore infer that the drug vehicle DMSO may have had deleterious effects on PGCs after specification. Furthermore, when we compared no-inhibition, continuous-inhibition and released-from-inhibition conditions (*Figure 8—figure supplement 1B*, compare Ctrl-Ctrl, GDC-GDC and GDC-Ctrl), PGC numbers did not vary significantly. These observations suggest that even though the initial PGC specification is Hh dependent, the PGC population can be dynamically replenished, potentially through cell proliferation. Additionally, we did not observe PGC migration defects in different combinations of GDC-0449 treatments. At 12 dpf we found that like control polyps, more than half of treated polyps still showed the expected PGC migration away from the clusters: 18 of 29 polyps in Ctrl-Ctrl; 18 of 30 polyps in Ctrl-GDC; 19 of 30 polyps in GDC-GDC; 16 of 30 polyps in GDC-Ctrl (*p* value of Chi-squared test comparing with Ctrl-Ctrl are 0.87, 0.92 and 0.5 respectively). Therefore, Hh signaling is not likely to be involved in PGC migration after the initial specification step.

## Hh signaling regulates endomesodermal patterning and PGC specification

To definitively test the requirements for Hh signaling in *Nematostella*, we next used an established CRISPR/Cas9 methodology to mutate *hh1* (*Ikmi et al., 2014*; *Kraus et al., 2016*; *He et al., 2018*). These efforts generated three F1 heterozygous lines carrying frame-shift mutations (*Figure 9*, *Figure 9—figure supplement 1*, *Figure 9—figure supplement 2*; see Materials and methods). In F2 progeny resulting from crosses between heterozygous F1 siblings, we observed developmental defects in primary polyps wherein body length was reduced by approximately 50% and the four primary tentacles failed to elongate and partially fused together (*Figure 9A–C'*, *Figure 9—figure*

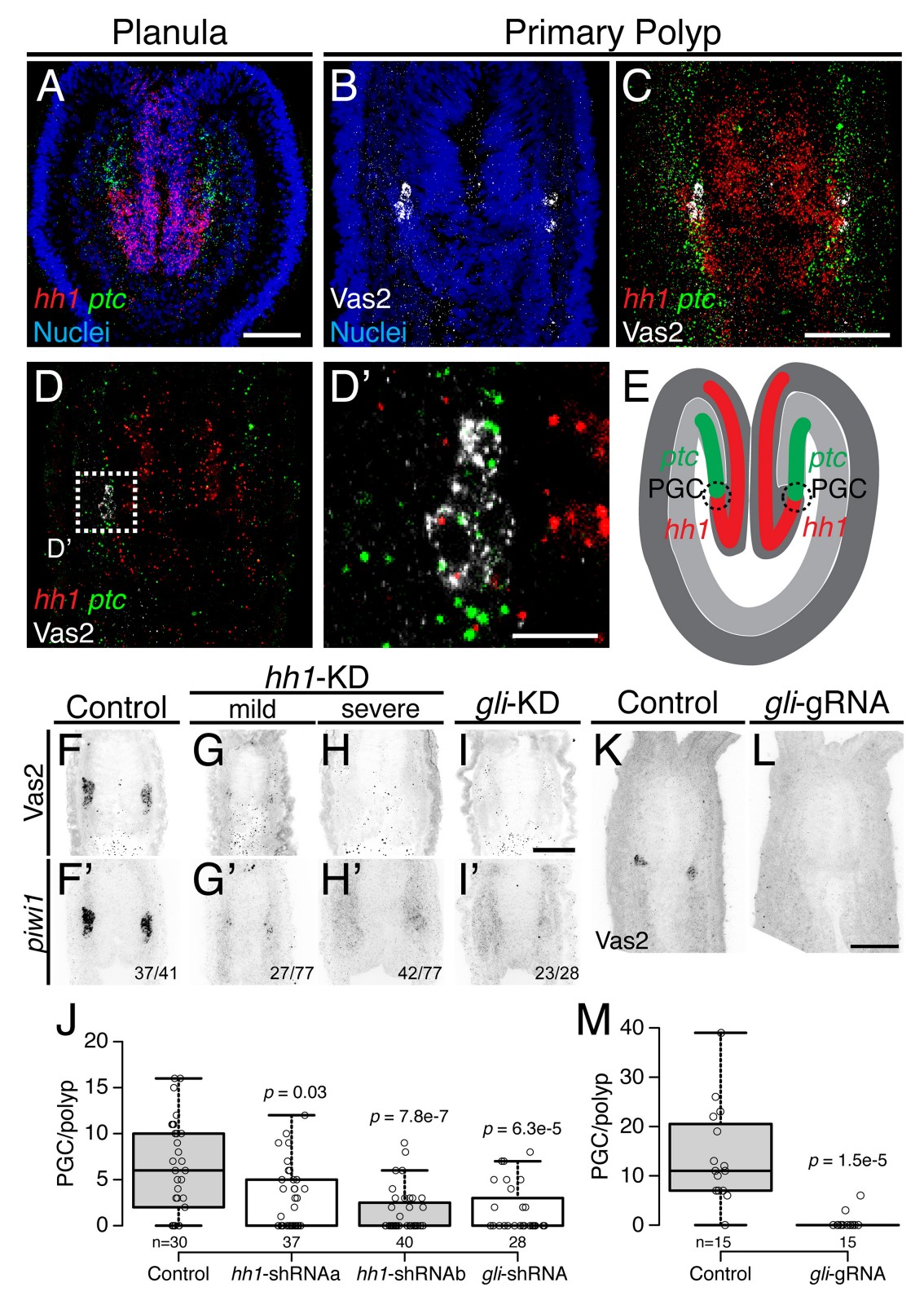

**Figure 7.** Hh signaling is required for *Nematostella* PGC formation. (**A**) Prior to PGC specification in planula larvae, *hh1* (*red*) and *ptc* (*green*) are expressed in pharyngeal ectoderm and endomesoderm, respectively. (**B–D'**) In primary polyps, PGC clusters (*white*) are specified within the *ptc* expression domain, neighboring the *hh1* domain. (**D**) Selected single focal plane of **C**. (**D'**) Enlarged view of boxed area in **D**, showing PGCs expressing *ptc* (*green*) and low *hh1* signal (*red*). (**E**) Diagram depicting the relative expression domains of *hh1* and *ptc* and PGC specification during

*Figure 7 continued on next page*

*Figure 7 continued*

metamorphosis. (F–I′) PGC formation on 8 dpf—indicated by Vas2 immunostaining and *piwi1* fluorescent in situ hybridization—is impaired by *hh1* or *gli* shRNA knockdown. (J) PGC numbers were significantly reduced following *hh1* or *gli* knockdown. *p* values of ANOVA tests were compared with control GFP-shRNA injection. (K–M) *gli* gRNAa, gRNAb and Cas9 protein injected embryos showed reduced PGC numbers in six dpf primary polyps. *p* value of ANOVA test was compared with control GFP-gRNA injection. Scale bar = 50 μm in A, C, I and L; 10 μm in D′. B-D′ are at the same scale; F–I′ are at the same scale; K and L are at the same scale.

The online version of this article includes the following source data and figure supplement(s) for figure 7:

**Source data 1.** PGC numbers in *hh1*- or *gli*-shRNA knockdown primary polyps.
**Source data 2.** PGC numbers in *gli*-gRNA injected primary polyps.
**Figure supplement 1.** The *Nematostella gli* locus and gRNA targeting sequences used in this work.

*supplement 2*). Consistent with a defect in Hh signal transduction, homozygous mutants expressed lower levels of *ptc*, a conserved Hh pathway target gene (*Figure 9D–E′*). Primary polyps homozygous for either *hh1* mutant allele developed primary mesentery-like endomesodermal septa; however, we did not observe Vas2, *piwi1* or *tudor* expressing PGC-like cluster cells (*Figure 9F–I′*).

Morphological analysis revealed abnormal internal tissue patterning in *hh1* homozygous mutants. In *wild-type* animals, the pharynx and primary septal filaments are separated from body wall ectoderm by intervening endomesodermal tissue (*Figure 10A–A′*). By contrast, in *hh1* mutants part of the pharynx and the primary septal filaments were in direct contact with the ectoderm (*Figure 10B–B′*). As a result, the eight segments of the larval body plan were abnormally segregated into groups of three and five segments by the pharynx (*Figure 10B*). These defects were not observed in *hh1* and *gli* shRNA knockdowns, suggesting that PGC formation may require a higher level of Hh signaling activity than endomesoderm patterning. The primary polyp-like *hh1* homozygous mutants passed through gastrulation, indicating that the pharynx and the endomesoderm likely formed a continuous epithelium. Consistent with the hypothesis that a pharyngeal signal induces PGC development, *hh1* mutants failed to develop PGCs even though the pharynx associated with the endomesoderm. Nevertheless, without more sophisticated genetic tools, we cannot rule out the possibility that PGC formation was indirectly perturbed by Hh-dependent endomesodermal patterning defects. In either case, we conclude that zygotic signaling activity is required for specification of the putative PGC clusters.

## PGC formation in *ptc* mutants may reflect a default Hh activation without the receptor

In bilaterian model systems, Ptc has been shown to serve as a receptor for the Hh ligand and to inhibit the pathway when the ligand is absent (*Johnson et al., 2000*). To further interrogate the mechanism of PGC specification in *Nematostella*, we generated four *ptc* heterozygous mutant lines (*Figure 10C–D*, *Figure 10—figure supplement 1*, *Figure 10—figure supplement 2*; see Materials and methods). Crosses between heterozygous siblings resulted in the expected 25% of homozygous progeny based on genotypic analysis, and these developed into abnormal mushroom-shaped polyps which lacked the four primary tentacles (*Figure 10C–D*, *Figure 10—figure supplement 2*). Detailed morphological examination and Vas2 immunofluorescence revealed that the *ptc* homozygous mutants developed a pharynx, eight endomesoderm mesenteries, and two PGC clusters (*Figure 10D–E′*). Combined with the requirement for *hh1*, *gli* and Smo activity during PGC specification, we propose that the presence of Hh ligand or absence of *ptc* activates the pathway and that zygotic Hh signaling provides permissive conditions for PGC formation within the pharyngeal domain of the *Nematostella* endomesoderm.

## Discussion

In this report, we confirm that *Nematostella* putative PGCs form in pharyngeal endomesoderm and provide evidence that these cells delaminate via EMT and migrate through the mesoglea to populate the eight gonad primordia. We also demonstrate that putative PGCs form between the expression domains of *hh1* and *ptc* and present evidence that Hh signaling is required for PGC specification but not PGC maintenance. Because Hh signaling transducers are only expressed zygotically (*Matus et al., 2008*; *Lotan et al., 2014*), these data indicate that *Nematostella* employs an

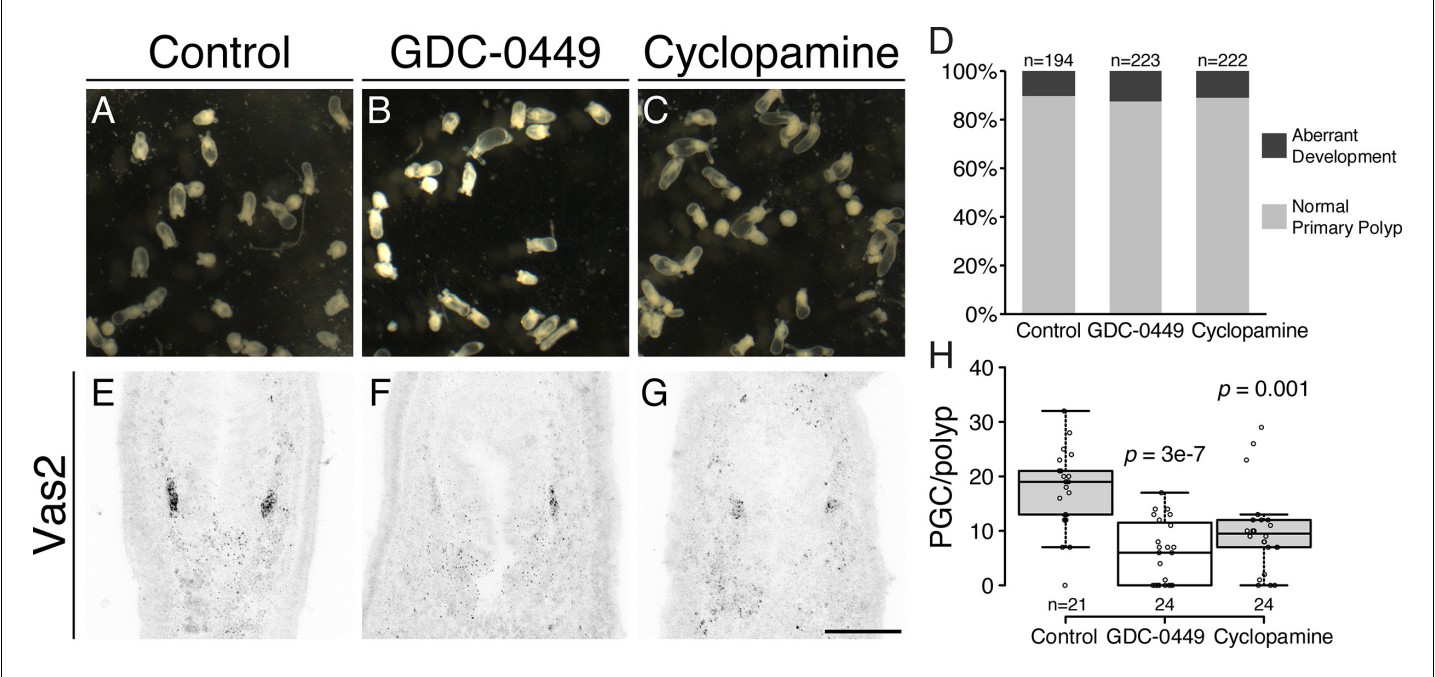

**Figure 8.** Inhibiting Hh signaling by GDC-0449 or Cyclopamine impairs PGC formation. (**A–D**) The majority of primary polyps did not show visible developmental defects after treatment with GDC-0449 or Cyclopamine from the gastrula stage onward. (**E–H**) Primary polyps treated with GDC-0449 or Cyclopamine from 1 to 8 dpf formed significantly fewer PGCs than controls. *p* values of ANOVA tests were compared with control treatment. Scale bar = 50 µm in **G**. **A**-**C** are at the same scale; **E**–**G** are at the same scale.

The online version of this article includes the following source data and figure supplement(s) for figure 8:

**Source data 1.** Quantification of normal primary polyps and aberrant development after GDC-0449 or Cyclopamine treatment.

**Source data 2.** PGC numbers in GDC-0449 or Cyclopamine treated primary polyps.

**Figure supplement 1.** The Hh signaling pathway does not affect PGC behaviors after specification.

**Figure supplement 1—source data 1.** PGC numbers in GDC-0449 treated 8 and 12 dpf primary polyps.

inductive mechanism to specify PGC fate, which is consistent with the proposed ancestral mechanism for metazoan PGC specification (*Extavour and Akam, 2003*). Considering the existence of maternally-derived perinuclear Vas2 granules and *vas1* and *nos2* transcripts (*Extavour et al., 2005*; *Praher et al., 2017*), it remains possible that maternally inherited germline determinants still play some essential roles in PGC specification and that zygotic Hh activity serves to augment their function. In this combined maternal-zygotic scenario, the mechanism of *Nematostella* PGC formation would not neatly fit within either inheritance or induction, but instead falls within the continuum between either extreme, similar to sea urchin PGCs where maternal and zygotic factors cooperate in PGC determination (*Nieuwkoop and Sutasurya, 1981*; *Voronina et al., 2008*; *Seervai and Wessel, 2013*).

We labeled the origins of *Nematostella* putative PGC by the expression of Vas2 and other conserved germline marker genes. However, the observations do not exclude the possibility that Vas2+ cells can give rise to other somatic lineages because these conserved germline marker genes are also expressed and functional in multipotent stem cells of many species, such as Hydra, planaria and *Platynereis dumerilii* (*Mochizuki et al., 2001*; *Reddien et al., 2005*; *Rebscher et al., 2007*; *Gustafson and Wessel, 2010*; *Wagner et al., 2012*). In line with the theory of a 'germline multipotency program' where PGCs and multipotent stem cells are sister cell types and the specification and maintence of multipotency rely on these genes (*Juliano et al., 2010*), it is possible that the Vas2 + cells of *Nematostella* polyps contribute to both germline and soma. Alternatively, because our observations were made from static images, multiple rounds of PGC specification from other origins may exist. Other yet-identified pluripotent stem cells may also contribute to the *Nematostella* germline even after the initial PGCs have been specified. Based on our current data, we can simply

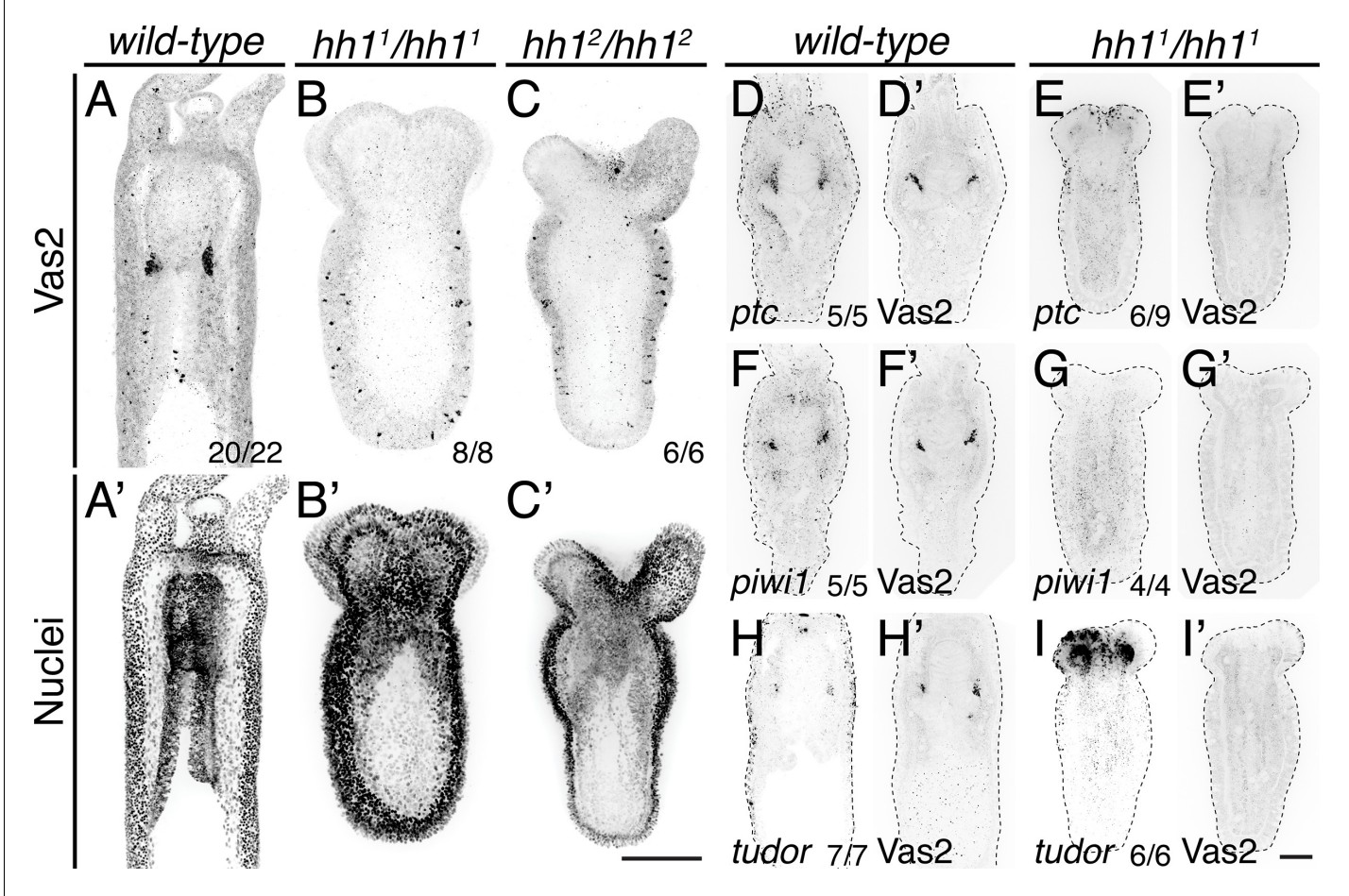

**Figure 9.** PGC clusters do not form in *hh1* mutants. (A–C') *hh1* homozygous mutant polyps exhibit a shorter body column and pronounced tentacle defects compared to *wild-type* siblings on 8 dpf. Additionally, no Vas2+ PGC clusters were detected in *hh1* homozygous knockout polyps. (D–I') FISH of *ptc*, *piwi1* or *tudor* were followed by Vas2 immunohistochemistry on 12 dpf *wild-type* and *hh1¹/hh1¹* mutants. (D–E') *hh1* mutant polyps show reduced *ptc* expression in the endomesoderm. (F–I') *hh1* mutant polyps lose PGC markers, including *piwi1*, *tudor* and Vas2. (A–C') are multiple-focal planes projections. (D–I') are from single-focal plane at the primary mesentery level. Scale bar = 50 μm in C' and I'; A–C' are at the same scale; D–I' are at the same scale.

The online version of this article includes the following figure supplement(s) for figure 9:

**Figure supplement 1.** The *Nematostella hh1* locus, mutant alleles and the deduced protein structure compared with Hh proteins of other species.
**Figure supplement 2.** The *hh³* mutant recapitulates the phenotype of *hh¹* and *hh²* homozygotes.

propose that PGCs originate from the Vas2+ cell clusters in primary polyps. With more advanced genetic tools, we anticipate that the *Nematostella* PGC lineage will be revealed by live tracing methods.

## Hh pathway activity in *ptc* mutants

In many bilaterian model organisms *ptc* is a transcriptional target of Hh signaling and serves as both a receptor and negative regulator of pathway activity (*Briscoe and Thérond, 2013*; *Bangs and Anderson, 2017*). We sought to functionally dissect Hh signaling in *Nematostella* and leveraged CRISPR/Cas9 mutagenesis to generate both *hh1* and *ptc* mutants. While *hh1* mutants lacked putative PGC cell clusters (*Figure 9*), to our surprise these cells formed properly in *ptc* mutant animals (*Figure 10*). This finding could be consistent with three possible scenarios: (1) The existence of residual receptor activity due to allele-specific effects or potential redundancy with an unannotated paralogue elsewhere in the genome; (2) An indirect or insufficient role for Hh in PGC specification; (3) A default repressive role for Ptc in the specification of pre-patterned PGC clusters. Because

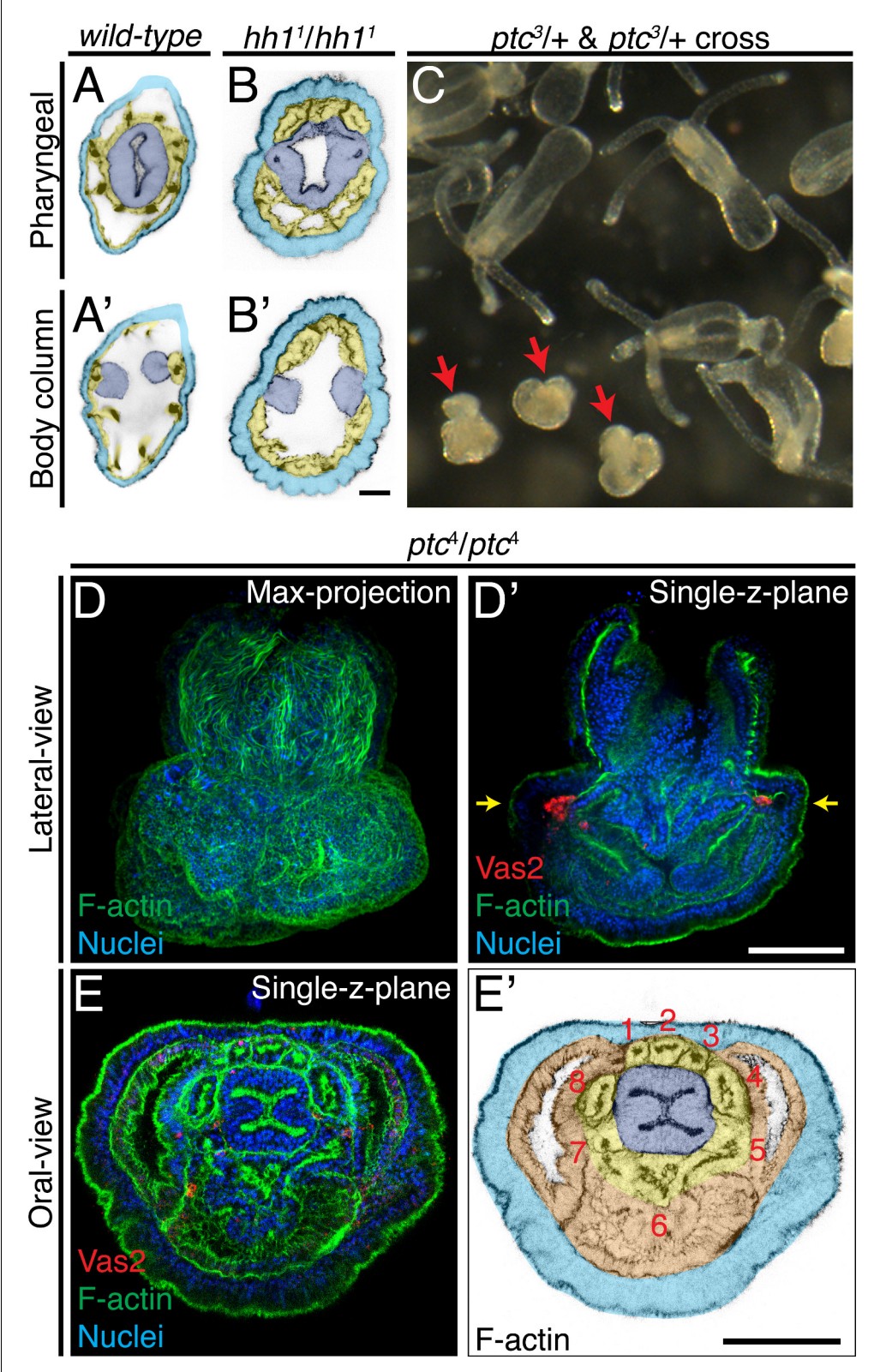

**Figure 10.** Patterning defects in *hh1* and *ptc* mutants. (A–B') In addition to loss of putative PGCs, *hh1* mutant polyps show endomesodermal patterning defects. Parts of the pharyngeal ectoderm and septal filaments (*navy blue*) abnormally contact the outer epidermis (*azure blue*), without endomesoderm tissue in between (*yellow*). These contacts segregated the normally contiguous eight endomesodermal segments into blocks of three and five segments along the directive axis. (C) Eighteen dpf F2 progeny from a cross between *ptc³/+* heterozygous siblings. The abnormal mushroom-shaped

*Figure 10 continued on next page*

*Figure 10 continued*

polyps are indicated by *red* arrows. (**D–D'**) At the primary polyp stage (12 dpf), homozygous *ptc* mutants lack the four primary tentacles and do not develop the normal polyp body plan. (**E–E'**) A single focal plane taken at the level indicated by *yellow* arrows in **D'**. Depite significant morphological defects, *ptc* mutant animals develop a pharynx (*navy blue*), eight endomesodermal segments (*yellow*), body wall endomesoderm (*orange*) and putative PGC clusters (labeled by Vas2 immunofluorescence in *red* in **D'**). Scale bar = 20 µm in **B'**; 50 µm in **D'** and **E'**. A-B' are at the same scale; D–D' are at the same scale; E–E' are at the same scale.

The online version of this article includes the following figure supplement(s) for figure 10:

**Figure supplement 1.** The *Nematostella ptc* locus, mutant alleles and the deduced protein structure compared to other species.

**Figure supplement 2.** Crosses between *ptc¹/+* and *ptc¹/+* or *ptc¹/+* and *ptc²/+* heterozygous siblings.

inhibiting Hh activity by disrupting either Smo or *gli* also disrupted PGC formation (*Figure 7I–M*; *Figure 8*), we suggest that Ptc most likely serves as a default inhibitor of Hh activity in *Nematostella*. Based on our combined data, we propose that the pharyngeal ectoderm releases Hh ligand to inhibit Ptc-dependent repression of PGC fate in the neighboring endomesoderm. This reasoning would suggest that the PGC clusters are pre-patterned by other yet-identified extracellular signals, and that the role of Hh activity may be to provide a spatial or temporal cue to trigger their maturation.

## Direct versus indirect roles for Hh activity in PGC specification

To our knowledge, Hh signaling has not been directly implicated in PGC specification in previous studies of established bilaterian systems. Nevertheless, as summarized in *Figure 11* and *Table 1* and *Table 2*, a requirement for Hh signaling during *Nematostella* PGC formation is supported by three lines of evidence: (1) *hh1* and *gli* shRNA knockdowns and *gli* CRISPR/Cas9 mutagenesis (*Figure 7*); (2) Smo inhibition assays (*Figure 8*); and (3) *hh1* mutants (*Figure 9*). In developing primary mesenteries, PGCs are specified in endomesoderm cells that lie in close proximity to the *hh1* expression domain in adjacent pharyngeal ectoderm (*Figure 7A–E*). Even in the absence of primary mesenteries in *Anthox6a* mutants and *Gbx* knockdown juveniles (*Figure 6*; *He et al., 2018*), PGCs still develop from endomesodermal cells in proximity to the pharyngeal ectoderm-derived septal filaments (*Steinmetz et al., 2017*). Interestingly, while *hh1* expression seems to be restricted to the pharyngeal ectoderm and septal filaments, we observed broad endomesodermal patterning defects in *hh1* mutants (*Figure 10B–B'*). This phenotype was not seen in either knockdown experiments or inhibitor assays where PGC specification was nevertheless inhibited (*Figure 7* and *Figure 8*). This could

**Table 1.** Primer sets for cloning probe templates into pPR-T4P (gene-specific regions are underlined).

| Target | Sequence | | Probe size (bp) |
|--------|----------|------|-----------------|
| *piwi1* | Forward | CATTACCATCCCGCGAGCCTACAACCAGGAGAG | 1327 |
| | Reverse | CCAATTCTACCCGCGTTGTGTTGATGCCCATAG | |
| *piwi2* | Forward | CATTACCATCCCGTGGGCGGTACTTCTACAACC | 1367 |
| | Reverse | CCAATTCTACCCGTGCCCTTGATAAGGAGCATC | |
| *vas2* | Forward | CATTACCATCCCGTGAAGGGTCTCCAATTCCTG | 1530 |
| | Reverse | CCAATTCTACCCGTGTGCAGATTACAGCCAAGG | |
| *tudor* | Forward | CATTACCATCCCGGAACCTACTTGCTTCCGCAG | 1450 |
| | Reverse | CCAATTCTACCCGACGACTCGGTGTTCCCATAG | |
| *twist* | Forward | CATTACCATCCCGAAATCTCGGTGTCGGTCTTG | 1020 |
| | Reverse | CCAATTCTACCCGTATCGCAGCTTTGCTTTCAG | |
| *hh1* | Forward | CATTACCATCCCGTTTCATTGGGAGCTAGTGGG | 1327 |
| | Reverse | CCAATTCTACCCGAAAGCGTGAATTGGGTCTTG | |
| *ptc* | Forward | CATTACCATCCCGGATGTGCGTGTGTGGGATAG | 1455 |
| | Reverse | CCAATTCTACCCGACCGCGAGGTAATTGAACAC | |

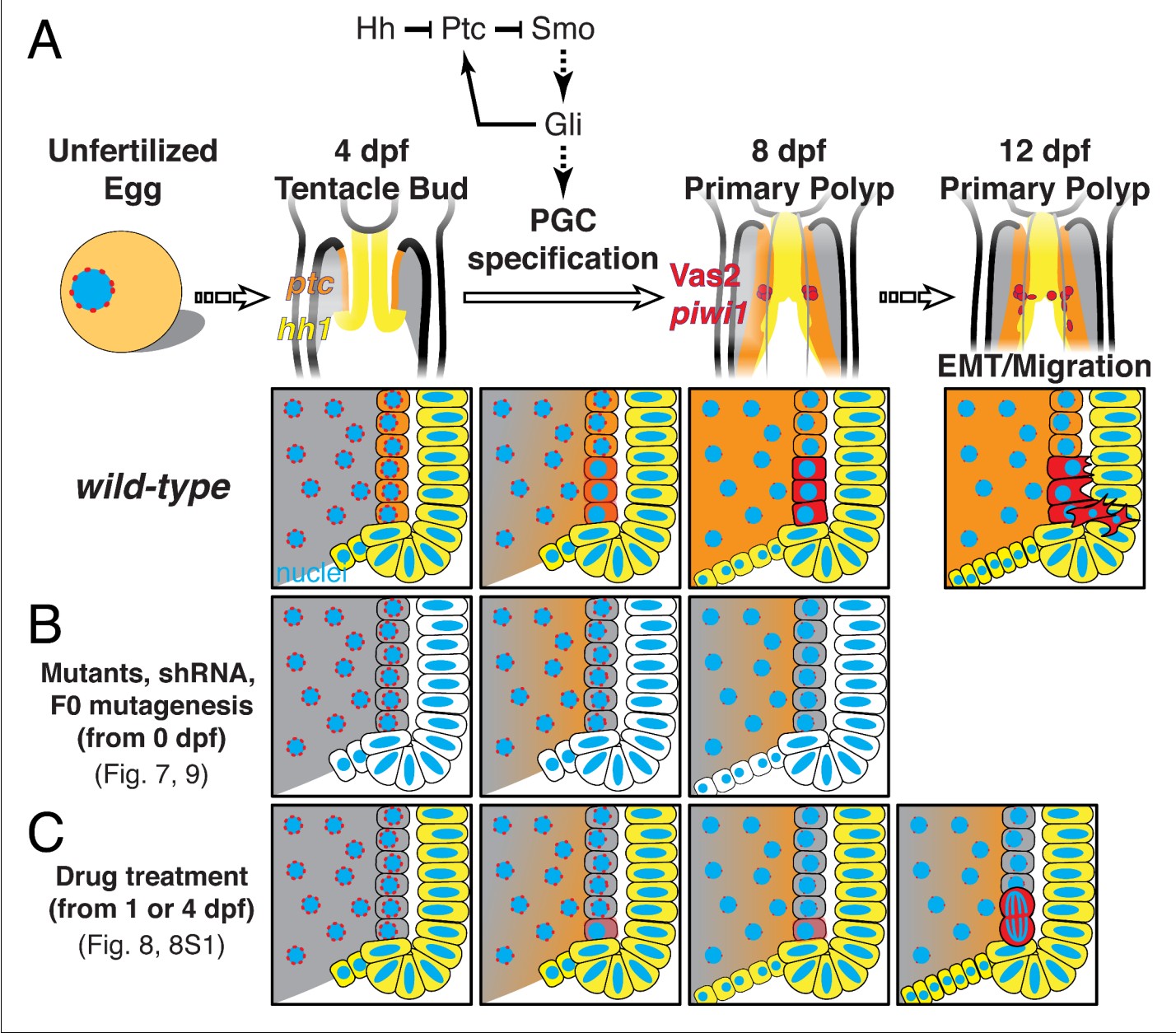

**Figure 11.** A model of *Nematostella* PGC specification and migration. (**A**) In *wild-type* 4 dpf tentacle bud larvae, *hh1* (*yellow*) and *ptc* (*orange*) are expressed in the pharyngeal ectoderm and endomesoderm, respectively. Between 4 to 8 dpf when larvae metamorphose to primary polyps, Hh1 signals to neighboring endomesodermal cells and specifies Vas2/*piwi1*-positive PGC clusters (*red*) in the primary mesenteries. Meanwhile, perinuclear Vas2 granules (*red* dots) within endodermal cells (*gray*) gradually diminish. After initial specification, the PGC clusters undergo EMT and migrate to gonad rudiments during the juvenile stage. (**B**) *hh1* KD, *gli* KD, *hh1* mutants and *gli* gRNA-Cas9 injected embryos develop reduced or absent PGCs clusters, indicating a requirement for Hh signaling in this process, whether direct or indirect. (**C**) Drug treatments inhibiting Smo activity between 4 to 8 dpf impair PGC specification. However, some polyps still form reduced numbers of PGCs at later time points, possibly due to compensatory PGC proliferation. Note that the reduced *ptc* expression depicted in **B** is supported by FISH data in *hh1* mutants but is presumptive in **C**.

suggest that the PGC defects in *hh1* mutants are a direct result of aberrant endomesodermal patterning. Consistent with this, preliminary attempts to broadly overexpress Hh by injecting *Ubiquitin>hh* plasmid did not result in the induction of any apparent ectopic PGCs (data not shown). Still, looking forward, genetic tools that allow the discrimination between cell autonomous and cell non-autonomous mechanisms will be required to definitively rule out whether PGC formation is directly or indirectly regulated by Hh signaling.

### Future perspectives

In this report, we provide an initial framework demonstrating an inductive PGC formation mechanism in *Nematostella vectensis*, a representative early-branching animal. To our knowledge, there is no direct evidence about the involvement of Hh signaling in PGC specification in other organisms. One report from Hara and Katow showed *hedgehog* expression in the small micromeres (PGC precursors) of the sea urchin, *Hemicentrotus pulcherrimus* (*Hara and Katow, 2005*; *Yajima and Wessel, 2011*). Functional interrogation of Hh signaling did not address sea urchin PGC formation but suggests the pathway patterns mesoderm and regulates left-right asymmetry (*Walton et al., 2009*; *Warner et al., 2016*). Because complete hedgehog protein homologs appeared in the cnidarian-bilaterian common ancestor and the pathway is involved in many facets of development (*Adamska et al., 2007*; *Ingham, 2001*; *King et al., 2008*), it is possible that the pathway may serve to distinguish germline and soma in other eumetazoans as well. Alternatively, the requirement for Hh signaling in *Nematostella* PGC formation could be a lineage-specific feature, which could be tested through a broad sampling of PGC development in diverse anthozoan cnidarians.

## Materials and methods

**Key resources table**

| Reagent type (species) or resource | Designation | Source or reference | Identifiers | Additional information |
|---|---|---|---|---|
| Gene (*Nematostella vectensis*) | *vas1* | GenBank | AY730696 | |
| Gene (*Nematostella vectensis*) | *vas2* | GenBank | AY730697 | |
| Gene (*Nematostella vectensis*) | *piwi1* | GenBank | MF683122 | |
| Gene (*Nematostella vectensis*) | *piwi2* | GenBank | MF683123 | |
| Gene (*Nematostella vectensis*) | *tudor* | GenBank | XM_032376221 | |
| Gene (*Nematostella vectensis*) | *twist* | GenBank | AY286509 | |
| Gene (*Nematostella vectensis*) | *hh1* | GenBank | EU162651 | |
| Gene (*Nematostella vectensis*) | *ptc* | GenBank | EU162650 | |
| Gene (*Nematostella vectensis*) | *gli* | GenBank | EU162649 | |
| Gene (*Nematostella vectensis*) | *Anthox6a* | GenBank | GQ240845 | |
| Gene (*Nematostella vectensis*) | *gbx* | GenBank | DQ500757 | |
| Genetic reagent (*Nematostella vectensis*) | *hh1*[1] | This paper | | See *Figure 9—figure supplement 1* |
| Genetic reagent (*Nematostella vectensis*) | *hh1*[2] | This paper | | See *Figure 9—figure supplement 1* |
| Genetic reagent (*Nematostella vectensis*) | *hh1*[3] | This paper | | See *Figure 9—figure supplement 1* |
| Genetic reagent (*Nematostella vectensis*) | *ptc*[1] | This paper | | See *Figure 10—figure supplement 1* |
| Genetic reagent (*Nematostella vectensis*) | *ptc*[2] | This paper | | See *Figure 10—figure supplement 1* |
| Genetic reagent (*Nematostella vectensis*) | *ptc*[3] | This paper | | See *Figure 10—figure supplement 1* |

*Continued on next page*

*Continued*

| Reagent type (species) or resource | Designation | Source or reference | Identifiers | Additional information |
|---|---|---|---|---|
| Genetic reagent (*Nematostella vectensis*) | *ptc*[4] | This paper | | See *Figure 10—figure supplement 1* |
| Peptide, recombinant protein | vas2 | This paper | | MCFKCQQTGHFARECP NESAAGENGDRP KPVTYVPPTPTEDEEEM FRSTIQQGINFEKYD QIEVLVSGNNPVRHINSFEEANLY EAFLNNVRKAQYKKPTPHHHHHH |
| Antibody | anti-vas2 (Rabbit polyclonal) | This paper | | IF (1:1000), stock: 0.4 mg/mL |
| Antibody | anti-α-Tubulin (mouse monoclonal) | Sigma-Aldrich | T9026 | IF (1:1000) |
| Antibody | anti-Phospho-Histone H3 (mouse monoclonal) | Sigma-Aldrich | 05–806 | IF (1:1000) |
| Antibody | anti-DIG-POD Fab fragments (Sheep polyclonal) | Sigma-Aldrich | 11207733910 | FISH (1:1000) |
| Antibody | anti-Fluorescein-POD Fab fragments (Sheep polyclonal) | Sigma-Aldrich | 11426346910 | FISH (1:1000) |
| Chemical compound, drug | GDC-0449 | Cayman Chemical | 13613 | 25 µM |
| Chemical compound, drug | Cyclopamine | Cayman Chemical | 11321 | 5 µM |
| Chemical compound, drug | Benzyl alcohol | Sigma-Aldrich | 305197 | |
| Chemical compound, drug | Benzyl benzoate | Sigma-Aldrich | B6630 | |
| Commercial assay or kit | Click-iT EdU Kit | Thermo Fisher Scientific | C10339 | |
| Commercial assay or kit | TSA Plus Cyanine 3 System | PerkinElmer | NEL744001KT | |
| Commercial assay or kit | ImProm-II Reverse Transcription System | Promega | A3800 | |
| Commercial assay or kit | T7 RNA polymerase Kit | Promega | P2077 | |
| Commercial assay or kit | AmpliScribe T7-Flash Transcription Kit | Lucigen | ASF3507 | |
| Software, algorithm | ImageJ | ImageJ (http://imagej.nih.gov/ij/) | RRID:SCR_003070 | |
| Software, algorithm | Imaris | Bitplane | RRID:SCR_007370 | 8.3 |
| Software, algorithm | BoxPlotR | BoxPlotR.shiny (https://github.com/VizWizard/BoxPlotR.shiny/blob/master/README.md) | RRID:SCR_015629 | |
| Other | Hoechst-34580 stain | Sigma-Aldrich | 63493 | 1 µg/mL |
| Other | SYBR Green I stain | Thermo Fisher Scientific | S7567 | 1:5000 |
| Other | Phalloidin stain | Thermo Fisher Scientific | A12379 | 1:200 |
| Other | DIG RNA labeling mix | Sigma-Aldrich | 11277073910 | |

*Continued on next page*

*Continued*

| Reagent type (species) or resource | Designation | Source or reference | Identifiers | Additional information |
|---|---|---|---|---|
| Other | Fluorescein RNA labeling mix | Sigma-Aldrich | 11685619910 | |
| Other | Cas9 protein with NLS | PNA Bio | CP02 | 500 ng/µL |

## Animal husbandry

Maintenance and spawning of *N. vectensis* followed previously established protocols (*Stefanik et al., 2013*). Embryos were cultured at 24°C incubator for consistent developmental staging.

## Generation of anti-Vas2 antibody

A His-tagged antigen for raising polyclonal Rabbit-anti-Vas2 antibody was designed, synthesized and purified by GenScript (Piscataway, NJ). The antigen sequence (MCFKCQQTGHFARECPNE SAAGENGDRPKPVTYVPPTPTEDEEEMFRSTIQQGINFEKYDQIEVLVSGNNPVRHINSFEEANLYEAF LNNVRKAQYKKPTP<u>HHHHHH</u>) partially encompasses the last zinc finger domain and the DEAD-like helicase domain of Vas2. In brief, the rabbit was immunized three times before checking the antibody titer, boosted once more and sacrificed for the whole serum. The serum was affinity purified, and the polyclonal antibody stock concentration is 0.4 mg/mL.

## Whole-mount immunofluorescence

Our immunohistochemical staining protocol generally followed *Genikhovich and Technau, 2009*, with the following modifications: samples were blocked in 5% goat serum diluted in PBS with 0.2% Triton X-100 (PTx) and 10% DMSO for increasing antibody penetration. Samples were then incubated in 1:1000 diluted stock of Rabbit-anti-Vas2 antibody, Mouse-anti-α-Tubulin (Sigma-Aldrich; St. Louis, MO; Cat. No. T9026) or Mouse-anti-Phospho-Histone H3 (Sigma-Aldrich; Cat. No. 05–806) in PTx with 0.1% DMSO and 5% goat serum at 4°C overnight. After six washes with PTx for at least 20 min each at room temperature, samples were incubated with Alexa Fluor Goat-anti-Rabbit or Goat-anti-Mouse secondary antibodies (Thermo Fisher Scientific; Waltham, MA) at 1:500 dilution in PTx with 5% goat serum at 4°C overnight. If desired, during secondary antibody incubation, samples were counter-stained for F-Actin with Phalloidin at 1:200 dilution (Thermo Fisher Scientific) and nuclei with either 1 µg/mL of Hoechst-34580 (Sigma-Aldrich; Cat. No. 63493), 140 µM DAPI, or 1:5000 diluted SYBR Green I (Thermo Fisher Scientific; Cat. No. S7567). After several washes, samples were serially immersed in a final solution of 80% of glycerol in PTx. Alternatively, we dehydrated samples in isopropanol and immersed in BABB (one part of benzyl alcohol and two parts of benzyl benzoate) to clear the tissue, allowing up to 400 µm imaging depth (*Wan et al., 2018*).

## EdU incorporation and Click-it reaction

To detect proliferating cells, juveniles were incubated in 800 µM EdU (Thermo Fisher Scientific) for 30 min and adults were incubated in 100 µM EdU for 12 hr together with artemia followed by fixation and dehydration with immunohistochemical staining protocol. The adult samples were then

**Table 2.** shRNA sequences.

| Target | shRNA name | Sequence |
|---|---|---|
| *hh1* | *hh1*-shRNAa | GGCTTGCTATAACACTGAT |
| *hh1* | *hh1*-shRNAb | GGCAGAGCTGTTGATATAA |
| *gli* | *gli*-shRNA | GAGAAGAGGGATTTCACAT |
| *Gbx* | *Gbx*-shRNAa | GCCAAGGTTAATAGATCCT |
| *Gbx* | *Gbx*-shRNAb | GGAAACGTGTACGATCACT |
| *eGFP* | *eGFP*-shRNA | GACGTAAACGGCCACAAGTT |

rehydrated and immersed in Tissue-Tek O.C.T. Compound (VWR) for cryosection at 10 μm intervals. After rehydration or washing out O.C.T. with PTx, whole-mount juveniles and tissue sections were incubated with Click-it reaction according to the manufacturer's instructions.

## Whole-mount fluorescent in situ hybridization (FISH)

To clone target genes, purified total RNA was reverse transcribed into cDNA by ImProm-II Reverse Transcription System (Promega; Madison, WI; Cat. No. A3800). Target gene fragments were first amplified from a mixed cDNA library of planula larva and primary polyps. Primers are listed in *Table 3*. We adopted a ligation-independent pPR-T4P cloning method (*Newmark et al., 2003*) to generate plasmids with probe templates and confirmed the positive clones by sequencing. We then PCR amplified the DNA template fragments using the AA18 (CCACCGGTTCCATGGCTAGC) and PR244 (GGCCCCAAGGGGTTATGTGG) primers, which flank the T7 promoter and the target gene sequence. After purifying DNA templates, we synthesized DIG-labeled RNA probes with the DIG RNA labeling mix (Sigma-Aldrich; Cat. No. 11277073910) and T7 RNA polymerase (Promega; Cat. No. P2077). Sample preparation, probe hybridization and signal detection followed established protocols (*Steinmetz et al., 2017*; *He et al., 2018*). The probe working concentration was 0.5 ng/μL for all genes.

For double FISH, we synthesized fluorescein-labeled RNA probes with the Fluorescein RNA labeling mix (Sigma-Aldrich; Cat. No. 11685619910) and hybridized together with a DIG-labeled probe of another gene. After detecting the first probe signal with TSA fluorescein reagent (PerkinElmer; Waltham, MA) and several washes with TNT buffer, we quenched peroxidase activity by incubating samples in 200 mM NaN$_3$/TNT for 1 hr. Samples were then washed six times with TNT for at least 20 min each and then subjected to second-round probe detection by either anti-DIG -POD Fab fragments (Sigma-Aldrich; Cat. No. 11207733910) or anti-Fluorescein-POD Fab fragments (Sigma-Aldrich; Cat. No. 11426346910).

## Short hairpin RNA knockdown

shRNA design, synthesis and delivery followed the protocol of *He et al., 2018* with the following modification: A reverse DNA primer containing the shRNA stem and linker sequence was annealed with a 20 nucleotide T7 promoter primer (TAATACGACTCACTATAGGG). The annealed, partially double stranded DNA directly served as template for in vitro transcription. We tested knockdown efficiency with shRNA produced by this modified method by targeting *β-catenin* and *dpp* shRNA, and found the same phenotypic penetrance as previously reported (*He et al., 2018*; *Karabulut et al., 2019*). To control for shRNA toxicity, we injected 1000 ng/μL eGFP shRNA and did not observe noticeable developmental defects. All shRNA working solutions were prepared at 1000 ng/μL and the sequences are listed in *Table 4*. By 8 dpf, primary polyps were fixed to assay PGC development.

## Generation of mutant lines by CRISPR/Cas9 mutagenesis

*hh1* and *ptc* mutant lines were generated using established methods (*Ikmi et al., 2014*; *Kraus et al., 2016*; *He et al., 2018*). In brief, to generate F0 founders, we co-injected 500 ng/μl of gRNA (sequences listed in *Table 5*) and 500 ng/μl of SpCas9 protein into unfertilized eggs. Mosaic F0 founders were then crossed with *wild-type* sperm or eggs to create a heterozygous F1 population. When the F1 polyps reached juvenile stage, we genotyped individual polyps by cutting tentacle

**Table 3.** gRNA sequence for CRISPR/Cas9 mutagenesis (PAM sequences are underlined).

| Target | gRNA name | Sequence |
|--------|-----------|----------|
| *hh1* | *hh1*-gRNAa | GGGAGCTAGTGGGAGACCAC<u>AGG</u> |
| *hh1* | *hh1*-gRNAb | GCTCATGAGGCGATCGGCAC<u>CGG</u> |
| *ptc* | *ptc*-gRNAa | GGAGGGGTTTGGAGCATCAC<u>AGG</u> |
| *ptc* | *ptc*-gRNAb | AGAGGTGAAGGCCAGGACAG<u>TGG</u> |
| *gli* | *gli*-gRNAa | GCGGCATGACCAGGAGGAGG<u>TGG</u> |
| *gli* | *gli*-gRNAb | AGTGAGGTGGCTGTGGATGG<u>TGG</u> |

**Table 4.** Summary of major observations during PGC development.

| PGC event | Developmental stage | Data |
|---|---|---|
| Specification on primary mesenteries | 4–8 dpf | *Figure 1B–D* |
| EMT and radial migration | juveniles with feeding | *Figure 2C–E', Figure 3B–B'* |
| Aboral migration to gonad rudiments | juveniles with feeding | *Figure 3C–D', Figure 4* |

samples; the resultant alleles are described in *Figure 9—figure supplement 1* and *Figure 10—figure supplement 1*. Heterozygous carriers of insertion/deletion-induced frame-shift alleles were crossed to generate homozygous mutants. The phenotypes and genotypes of the F2 population followed Mendelian inheritance and were subjected to further analysis. In progeny resulting from a $hh1^1/+$ cross, the observed phenotypic ratio of *wild-type* and mutant primary polyps was 948:343, close to the expected Mendelian ratio. Progeny from heterozygous crosses were also randomly genotyped and confirmed to follow the expected 1:2:1 ratio ($+/+$: $hh1^1/+$: $hh1^1/hh1^1$ = 6:14:8 and $+/+$: $hh1^2/+$: $hh1^2/hh1^2$ = 7:16:6). A similar strategy was used to analyze *ptc* mutants. *ptc* mutant genotypes also followed Mendelian segregation ($+/+$: $ptc^3/+$: $ptc^3/ptc^3$ = 5:17:8). These results suggest the phenotypes observed in the *hh1* and *ptc* mutant lines result from single locus mutations.

We tried to generate *gli* mutant lines by co-injecting two gRNAs (500 ng/µl each; *Table 1* and *Figure 7—figure supplement 1*) with 500 ng/µl of SpCas9 protein into unfertilized eggs. However, the F0 founders were >90% lethal at juvenile stage and the survivors were sterile. Therefore, we analyzed PGC formation in the F0 generation and as shown in *Figure 7K–M*.

## Inhibitor treatments

GDC-0449 (Vismodegib, Cayman Chemical; Ann Arbor, MI; Cat. No. 13613) and Cyclopamine (Cayman Chemical; Cat. No. 11321) were diluted in DMSO as 50 mM and 10 mM stocks, respectively.

**Table 5.** Summary of experimental designs.

| Experiment | Treatment duration | Data |
|---|---|---|
| Verify PGC specification in *hh1* and *gli* shRNA knockdown | 0–8 dpf | *Figure 7F–J* |
| Verify PGC specification in *gli* mutagenesis | 0–6 dpf | *Figure 7K–M* |
| Verify PGC specification in Smo inhibition by GDC-0449 and Cyclopamine | 1–8 dpf | *Figure 8* |
| Test Hh pathway on specification and post-specification by GDC-0449 temporal treatments | 4–8, 4–12 and 8–12 dpf | *Figure 8—figure supplement 1* |
| Verify PGC specification in $hh1^1$ and $hh1^2$ mutants | 0–8 dpf | *Figure 9A–C'* |
| Verify *ptc* and PGC marker expression in $hh1^1$ mutant | 0–12 dpf | *Figure 9D–I'* |
| Verify PGC specification in $hh1^3$ mutant | 0–12 dpf | *Figure 9—figure supplement 2* |
| Demonstrate patterning defects in $hh1^1$ mutant | 0–12 dpf | *Figure 10A–B'* |
| Demonstrate the mutant morphology of $ptc^3$ mutant | 0–18 dpf | *Figure 10C* |
| Verify PGC specification and demonstrate the mutant morphology of $ptc^4$ mutant | 0–12 dpf | *Figure 10D–E'* |
| Demonstrate the mutant morphology of $ptc^1$ and $ptc^2$ mutants | 0–8 dpf | *Figure 10—figure supplement 2* |

These stocks were diluted 1:2000 in 12 ppt filtered artificial sea water (FSW) to generate working solution: 25 µM GDC-0449 and 5 µM Cyclopamine. 1:2000 diluted 100% DMSO (final 0.05%) was applied as a control. All treatments were protected from light and replaced with fresh working solutions every day.

### Imaging and quantification

For confocal imaging, we used a Leica TCS Sp5 Confocal Laser Scanning Microscope or a Nikon 3PO Spinning Disk Confocal System. Bright field images were acquired using a Leica MZ 16 F stereoscope equipped with QICAM Fast 1394 camera (Qimaging; Surrey, BC, Canada). The brightness and contrast of images were adjusted by Fiji, and the PGC number of individual polyp was manually quantified by the Cell Counter Macro or automatically by blurring and masking the Vasa2 signal to find the cluster and 3D peak finding of the DAPI nuclei within the cluster (https://github.com/jouyun/pub-2020elife; *Chen, 2020*; copy archived at https://github.com/elifesciences-publications/pub-2020elife). Serial z section images of *N. vectensis* pharyngeal structures were reconstructed as a 3D movie (*Video 1*) by Imaris 8.3 (Bitplane, Concord, MA). Box plots were generated by BoxPlotR (*Spitzer et al., 2014*). Statistic analysis was done by one-way analysis of variance (ANOVA). We recognize significant difference when the *p*-value is less than 0.05. Figures of this report were generated using Adobe Illustrator 2019.

## Acknowledgements

The authors would like to thank Gibson lab members for their suggestions and critical readings of the manuscript. Additionally, we thank the Stowers Institute Core Technology Centers, in particular Aquatics for *Nematostella* husbandry, Molecular Biology for sequencing and genotyping, and Histology for help with histological sectioning. We also appreciate insightful comments and suggestions from the reviewers and editors of *eLife*. This work was funded in its entirety by the Stowers Institute for Medical Research.

## Additional information

### Funding

| Funder | Author |
|---|---|
| Stowers Institute for Medical Research | Matthew C Gibson |

The funders had no role in study design, data collection and interpretation, or the decision to submit the work for publication.

### Author contributions

Cheng-Yi Chen, Conceptualization, Data curation, Formal analysis, Visualization, Writing - original draft, Writing - review and editing; Sean A McKinney, Data curation, Formal analysis, Methodology, Writing - review and editing; Lacey R Ellington, Data curation, Methodology, Writing - review and editing; Matthew C Gibson, Conceptualization, Supervision, Funding acquisition, Writing - original draft, Writing - review and editing

### Author ORCIDs

Cheng-Yi Chen (iD) https://orcid.org/0000-0002-0520-346X
Sean A McKinney (iD) https://orcid.org/0000-0002-9740-6247
Lacey R Ellington (iD) https://orcid.org/0000-0003-0872-5455
Matthew C Gibson (iD) https://orcid.org/0000-0001-5588-8842

### Decision letter and Author response

Decision letter https://doi.org/10.7554/eLife.54573.sa1
Author response https://doi.org/10.7554/eLife.54573.sa2

## Additional files

### Supplementary files
• Transparent reporting form

### Data availability

Original data underlying this manuscript can be accessed from the Stowers Original Data Repository at http://www.stowers.org/research/publications/libpb-1492.

The following dataset was generated:

| Author(s) | Year | Dataset title | Dataset URL | Database and Identifier |
|---|---|---|---|---|
| Chen CY, McKinney SA, Ellington LR, Gibson MC | 2020 | Hedgehog signaling is required for endomesodermal patterning and germ cell development in the sea anemone Nematostella vectensis | http://www.stowers.org/research/publications/libpb-1492 | Stowers Original Data Repository, libpb-1492 |

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
