## [Decision Letter]

**Acceptance summary:**

This paper examines the mechanisms by which primordial germ cells (the cells that ultimately produce gametes) are formed in the sea anemone, *Nematostella vectensis*. Using a combination of genetic and pharmacological approaches the authors show that Hedgehog signaling, an evolutionarily conserved cell-cell communication pathway, is required for germ cell specification in this animal. This work expands our knowledge of germ cell specification mechanisms across the animal kingdom and will help us understand the evolutionary origins of this critical cell type.

**Decision letter after peer review:**

Thank you for submitting your article "Hedgehog signaling is required for endomesodermal patterning and germ cell development in *Nematostella vectensis*" for consideration by *eLife*. Your article has been reviewed by three peer reviewers, and the evaluation has been overseen by a Reviewing Editor and Patricia Wittkopp as the Senior Editor. The following individual involved in review of your submission has agreed to reveal their identity: Ulrich Technau (Reviewer #3).

The reviewers have discussed the reviews with one another and the Reviewing Editor has drafted this decision to help you prepare a revised submission.

Summary:

Chen et al. address the developmental origin of the primordial germ cells (PGCs) in the sea anemone *Nematostella vectensis*. They show that putative PGCs form in the pharyngeal endomesoderm of the first two mesenteries, at the border to the pharyngeal ectoderm. They propose that these cells delaminate via EMT and migrate through the mesoglea to populate the eight gonad primordia during the juvenile-to-adult stage transition. The region of PGC formation expresses the Hedgehog receptor patched (*ptc*), while the adjacent tissue expresses the hedgehog ligand *hh1*. Using CRISPR-based knockouts, knockdowns, and chemical inhibition of the Hh signaling pathway they show that the initial specification but not maintenance of PGC requires Hh signaling. The authors conclude that these data support inductive specification as the ancestral mechanism of PGC formation.

Essential revisions:

1) Since this work is the first to characterize PGC specification in *Nematostella* with such detail, the authors should carefully and comprehensively describe with quantification what they observe at each developmental stage in a systematic way. How many PGCs are there within each cluster at successive developmental timepoints? The authors attribute the increase in PGCs to proliferation (even though additional specification has not been ruled out), based on an experiment in which EdU incorporation occurs in Vas2+ cells (Figure 4—figure supplement 1H); however, the authors make no mention of this figure in the manuscript and fail to explain how they carried out the EdU experiment in the manuscript text. The EdU experiment is not detailed in the Materials and methods section either.

2) Assuming that the appearance of the two Vasa-positive cell clusters at the pharyngeal endomesoderm in the primary polyps is the moment of PGC specification, it would be helpful to clarify the exact timepoint this specification occurs (in dpf) relative to the time points that are tested in the knockdown/ knockout experiments. The reader assumes that PGCs do not form in the Hh/gli siRNA and CRISPR animals and that there is not an earlier time point at which PGCs initially form and then are lost. Thus, the authors suggest that Hh signaling is involved in specification. It would be helpful for the authors to clarify this point; providing a comprehensive model at the end that includes the timing could help.

3) The severe morphological phenotype of the patched mutant contrasts with the presence of the two clusters of PGCs, while no PGCs form in *Hh1* mutant or Gli3 knockdown embryos (Figure 8). Assuming that Patched acts as an inhibitor of Hh signaling, should the lack of *ptc* not lead to an ectopic activation of the Hh pathway and therefore ectopic formation of PGCs? How is the severe morphological phenotype explained, without an effect on PGC formation?

4) To distinguish between a direct versus indirect role for Hh signaling: if Hh activity acts directly to induce PGCs, then elevated or ectopic levels of Hh might induce supernumerary PGCs (or PGCs in ectopic places) – do Hh pathway agonists induce PGC formation? Is it possible to do gain-of function experiments? Is the effect of Hh signaling only restricted to a tissue of competence at the border of endoderm and ectoderm? What is the expression of Smoothened? Is it co-expressed with *ptc*?

5) The authors postulate that Hh signaling is crucial for the early induction of PGCs in the primary mesenteries, which then migrate circumferentially and aborally to the gonad region in all eight mesenteries. A crucial question is whether the formation of PGCs and later gametes is fully dependent on the pharyngeal tissue or whether it can be restored from other parts of the body. A straightforward experiment to test this would be to bisect the young polyps before the PGCs from the pharynx have reached the aboral region and check whether the regenerating polyps of the aboral half remain sterile or whether they can restore the Vas2-positive germ cells from other somatic cells and are fertile once they have reached sexual maturity.

6) The authors state that "upon feeding Vas2-expressing putative PGCs appeared to delaminate from the epidermis into the underlying mesogloea. Delaminated Vas2+ cells displayed a fibroblast-like morphology." While EMT followed by migration is a plausible explanation for the observation, it is still difficult to distinguish between new expression and migration of delaminated cells. Is this process continuing in adult polyps? Please comment.

7) The authors state in subsection “Evidence that PGCs form in primary polyps and migrate to gonad rudiments” that the PGCs migrate away from the high BMP activity domain along the directive axis. Is there evidence that the cells do this in response to the BMP gradient? Is there actually evidence that the BMP gradient is still present at this stage?

8) As the authors bring up in the Discussion, and as suggested by Nieuwkoop and Sutasurya in their 1981 monograph, germline determination mechanisms may be better described as occurring over a continuum between maternal specification and zygotic induction. *Nematostella* is a great example of this because germline factors (e.g. Vasa protein) are maternally inherited, but zygotic mechanisms are also clearly at play. The authors should consider reframing the manuscript to better reflect such a continuum of mechanisms. Rather than stating in the Introduction that the aim is to place *Nematostella* into one of the two bins, they should consider writing a more nuanced description of germline determination occurring over a continuum. Going beyond descriptive studies and understanding functionally how the germline is specified in a wider variety of organisms is required to understand which specific mechanisms (e.g. Hedgehog signaling?) are ancestral and which are derived. This is a strength of this study and could be emphasized more from the start, while de-emphasizing the placement of *Nematostella* into a specific bin.

9) Reviewers noted the difficulty of integrating the different results from gene knockdown, knockouts, and pharmacological manipulations into a coherent picture. This difficulty may be partly due to the order in which the data is presented and lack of details in the figure legends (see below). A paragraph (and/or a model at the end) integrating the results from all functional experiments as well as a clear indication of time points tested in each experiment would be really helpful. For example, it appears that the data shown in Figure 6—figure supplement 1 demonstrate that when the drug treatment is done between 4 and 8 days, there is a reduction in PGCs, but that the numbers recover to wild type levels at day 12. The Materials and methods section states that the shRNA-treated samples were tested at 8 days, but not 12 (this should be made clear in the figure legend for Figure 5).

10) Readability of the manuscript can be improved by presenting the figures in a logical order (instead of presenting figures and supplements out of order or not at all). For example, the authors do not refer to Figure 1A-D in the text and only refer to Figure 1E for the first time after Figures 2, 3, and 4! The authors should refer to each figure and panel sequentially, which may mean adding additional figures or rearranging current ones.

---

## [Author Response]

Essential revisions:1) Since this work is the first to characterize PGC specification in Nematostella with such detail, the authors should carefully and comprehensively describe with quantification what they observe at each developmental stage in a systematic way. How many PGCs are there within each cluster at successive developmental timepoints?

To address this concern, we have now quantified PGCs in each cluster after the cell lineage is established in primary polyps. This data is now included as Figure 1F and Figure 1—figure supplement 2.

The authors attribute the increase in PGCs to proliferation (even though additional specification has not been ruled out), based on an experiment in which EdU incorporation occurs in Vas2+ cells (Figure 4—figure supplement 1H); however, the authors make no mention of this figure in the manuscript and fail to explain how they carried out the EdU experiment in the manuscript text. The EdU experiment is not detailed in the Materials and methods section either.

To address this concern, we have now added a description of the EdU+/Vas2+ cells and Phospho-Histone H3+/Vas2+ cells in juvenile polyps and in adult sections of the main text as well as Figure 3—figure supplement 1. In addition, we have described Phospho-Histone H3 and EdU incorporation/detection in Materials and methods section.

2) Assuming that the appearance of the two Vasa-positive cell clusters at the pharyngeal endomesoderm in the primary polyps is the moment of PGC specification, it would be helpful to clarify the exact timepoint this specification occurs (in dpf) relative to the time points that are tested in the knockdown/ knockout experiments. The reader assumes that PGCs do not form in the Hh/gli siRNA and CRISPR animals and that there is not an earlier time point at which PGCs initially form and then are lost. Thus, the authors suggest that Hh signaling is involved in specification. It would be helpful for the authors to clarify this point; providing a comprehensive model at the end that includes the timing could help.

To address this concern, we have specified the exact time point of PGC specification of wildtype larva between 4-8 dpf as well as the tested time points in each figure legend. In addition, as suggested, we have now summarized our observations and experiments in Table 4 and 5 and proposed a PGC specification model in Figure 9.

In shRNA knockdown and mutant analysis experiments, we verified PGC specification at 8 dpf (except for 6 dpf for *gli* mutagenesis) for the following reasons: (1) Enhanced Vas2 expression in PGC clusters becomes more easily distinguished from the perinuclear signal in endomesodermal cells after 8 dpf, eliminating a possible source of error; (2) At 8 dpf, complete metamorphosis and consistently sized primary polyps are expected in >95% of individuals. Although we cannot rule out the possibility that PGCs formed at an earlier time point and were then lost, we think examination at 8 dpf fulfills the criteria for normal PGC specification and allows clear quantification of PGC numbers.

3) The severe morphological phenotype of the patched mutant contrasts with the presence of the two clusters of PGCs, while no PGCs form in Hh1 mutant or Gli3 knockdown embryos (Figure 8). Assuming that Patched acts as an inhibitor of Hh signaling, should the lack of ptc not lead to an ectopic activation of the Hh pathway and therefore ectopic formation of PGCs? How is the severe morphological phenotype explained, without an effect on PGC formation?

Considering Ptc serves as a negative regulator of the Hh pathway in other systems, we were also surprised that we did not observe ectopic PGC formation in *ptc* mutant animals. As discussed in subsection “Hh pathway activity in *ptc* mutants”, this suggest that Hh signaling is necessary but not sufficient for PGC specification. Other signaling molecules may play direct roles.

One possible hypothesis to explain these observations is that PGC specification requires maximal levels of signaling only achieved in the presence of Hh ligand. In *Drosophila*, the downstream transcription factor Ci exists in activator, full-length and suppressor forms which exhibit high-to-low transcriptional activities and activate different sets of genes (PMID: 15104233). Hh ligand promotes full-length Ci transformation into its activator form (PMID: 10102270 and PMID: 10862750), while Ptc promotes proteolysis of full-length Ci into a repressor form (Johnson *et al.,* 2020). Ci proteolysis is impaired in *ptc* mutants, leading to accumulation of full-length Ci. However, full-length Ci still requires Hh to be transformed into the activator form (PMID: 10102270). Therefore, higher levels of signaling may be achieved in wild-type animals than in *ptc* mutants. An in vitro assay also suggest that Hh ligand induces higher pathway activity than loss of *ptc* and that this phenomenon is Smo dependent (PMID: 30144507). Based on this information, we can speculate that Hh ligand enhances gli activation to support PGC specification in wildtype *Nematostella*, while in *ptc* mutants, gli remains full-length and may still be sufficient to activate PGC specification genes.

4) To distinguish between a direct versus indirect role for Hh signaling: if Hh activity acts directly to induce PGCs, then elevated or ectopic levels of Hh might induce supernumerary PGCs (or PGCs in ectopic places) – do Hh pathway agonists induce PGC formation? Is it possible to do gain-of function experiments? Is the effect of Hh signaling only restricted to a tissue of competence at the border of endoderm and ectoderm? What is the expression of Smoothened? Is it co-expressed with ptc?

We attempted overexpressing hh::2A::mScarlet fusion protein by injecting ubiquitin promoter-driven DNA constructs (see Author response image 1). We hypothesized that ectopic expression of *hh* would induce PGC formation in competent tissue domains, presumably mesendoderm. Except for normal PGC clusters (arrows), we did not observe ectopic vas2+ cells. The data suggest that either hh alone is not sufficient for specifying PGCs or hh from the DNA construct does not act like endogenous protein. Non-specific vas2 signals in pharynx and mesentery filaments were resulted from artifacts (*yellow* arrowheads).

**Author response image 1. sa2fig1:** 

We have not tested the expression pattern of the two Smoothened homologs. From Matus *et al.,* 2008, *gli* is co-expressed with *ptc* in the endomesoderm.

5) The authors postulate that Hh signaling is crucial for the early induction of PGCs in the primary mesenteries, which then migrate circumferentially and aborally to the gonad region in all eight mesenteries. A crucial question is whether the formation of PGCs and later gametes is fully dependent on the pharyngeal tissue or whether it can be restored from other parts of the body. A straightforward experiment to test this would be to bisect the young polyps before the PGCs from the pharynx have reached the aboral region and check whether the regenerating polyps of the aboral half remain sterile or whether they can restore the Vas2-positive germ cells from other somatic cells and are fertile once they have reached sexual maturity.

To address this point we bisected polyps at 8 days post-fertilization and tested whether aboral body fragments lacking pharyngeal tissue were able to regenerate Vas2+ clusters (see Author response image 2). In oral control fragments with intact pharyngeal tissue, Vas2+ clusters were present after amputation in 19/20 cases. In contrast, 83% of fully regenerated aboral fragments failed to regenerate Vas2+ clusters (n=12). We propose the following non-mutually exclusive explanations for these observations: (1) Regenerating aboral fragments did not have sufficient time to re-initiate Vas2 cluster formation; (2) The regenerated pharynx did not express sufficient Hh for re-specification of PGCs; and (3) Following bisection, the remaining aboral tissue was not competent to be specified as PGCs. Unfortunately, due to the outbreak of COVID-19, we were not able to test the long-term fertility of animals regenerated from aboral tissue.

6) The authors state that "upon feeding Vas2-expressing putative PGCs appeared to delaminate from the epidermis into the underlying mesogloea. Delaminated Vas2+ cells displayed a fibroblast-like morphology." While EMT followed by migration is a plausible explanation for the observation, it is still difficult to distinguish between new expression and migration of delaminated cells. Is this process continuing in adult polyps? Please comment.

We agree that without live imaging it is difficult to distinguish between EMT/migration and new expression of Vas2-expressing cells. To address this concern in the revised manuscript, we now propose alternative sources of PGCs in the Discussion section of the revised manuscript.

With regard to the continuation of the process in adult polyps, while we observed EMT/delamination of Vas2+ cells at the level of the lower pharynx in 8-tentacle stage juveniles, this cell population was not observed in more mature 12-tentacle stage adults. These observations require further confirmation but are most consistent with EMT/delamination occurring during a specific developmental window and not continuously in adults.

7) The authors state in subsection “Evidence that PGCs form in primary polyps and migrate to gonad rudiments” that the PGCs migrate away from the high BMP activity domain along the directive axis. Is there evidence that the cells do this in response to the BMP gradient? Is there actually evidence that the BMP gradient is still present at this stage?

This was an oversight in the original manuscript and we therefore modified the text accordingly: “The direction of this initial migration was toward segment s1, suggesting the existence of attractive/repulsive signals for migratory PGCs.”

8) As the authors bring up in the Discussion, and as suggested by Nieuwkoop and Sutasurya in their 1981 monograph, germline determination mechanisms may be better described as occurring over a continuum between maternal specification and zygotic induction. Nematostella is a great example of this because germline factors (e.g. Vasa protein) are maternally inherited, but zygotic mechanisms are also clearly at play. The authors should consider reframing the manuscript to better reflect such a continuum of mechanisms. Rather than stating in the Introduction that the aim is to place Nematostella into one of the two bins, they should consider writing a more nuanced description of germline determination occurring over a continuum. Going beyond descriptive studies and understanding functionally how the germline is specified in a wider variety of organisms is required to understand which specific mechanisms (e.g. Hedgehog signaling?) are ancestral and which are derived. This is a strength of this study and could be emphasized more from the start, while de-emphasizing the placement of Nematostella into a specific bin.

We agree with this suggestion and have expanded on the idea of a continuum of PGC specification mechanisms in the Introduction and reframed our main question around understanding mechanisms of germline specification instead of placing *Nematostella* into either bin.

9) Reviewers noted the difficulty of integrating the different results from gene knockdown, knockouts, and pharmacological manipulations into a coherent picture. This difficulty may be partly due to the order in which the data is presented and lack of details in the figure legends (see below). A paragraph (and/or a model at the end) integrating the results from all functional experiments as well as a clear indication of time points tested in each experiment would be really helpful. For example, it appears that the data shown in Figure 6—figure supplement 1 demonstrate that when the drug treatment is done between 4 and 8 days, there is a reduction in PGCs, but that the numbers recover to wild type levels at day 12. The Materials and methods section states that the shRNA-treated samples were tested at 8 days, but not 12 (this should be made clear in the figure legend for Figure 5).

As suggested, we summarized our observations and experiments in Table 4 and 5 and now propose a PGC specification model in Figure 9. Also, we described the treatment and PGC examination time in each figure legend.

10) Readability of the manuscript can be improved by presenting the figures in a logical order (instead of presenting figures and supplements out of order or not at all). For example, the authors do not refer to Figure 1A-D in the text and only refer to Figure 1E for the first time after Figures 2, 3, and 4! The authors should refer to each figure and panel sequentially, which may mean adding additional figures or rearranging current ones.

We apologize for this mistake and the manuscript has been revised accordingly. We now refer to Figure 1A-E’’ before Figure 2 in the revised manuscript.